# stClinic dissects clinically relevant niches by integrating spatial multi-slice multi-omics data in dynamic graphs

Chunman Zuo [1,2,3,4] ✉, Junjie Xia[2], Yupeng Xu[2], Ying Xu[5], Pingting Gao[3], Jing Zhang[6], Yan Wang [7] & Luonan Chen[8,9,10,11,12] ✉

Spatial multi-slice multi-omics (SMSMO) integration has transformed our understanding of cellular niches, particularly in tumors. However, challenges like data scale and diversity, disease heterogeneity, and limited sample population size, impede the derivation of clinical insights. Here, we propose stClinic, a dynamic graph model that integrates SMSMO and phenotype data to uncover clinically relevant niches. stClinic aggregates information from evolving neighboring nodes with similar-profiles across slices, aided by a Mixture-of-Gaussians prior on latent features. Furthermore, stClinic directly links niches to clinical manifestations by characterizing each slice with attention-based geometric statistical measures, relative to the population. In cancer studies, stClinic uses survival time to assess niche malignancy, identifying aggressive niches enriched with tumor-associated macrophages, alongside favorable prognostic niches abundant in B and plasma cells. Additionally, stClinic identifies a niche abundant in *SPP1+ MTRNR2L12+* myeloid cells and cancer-associated fibroblasts driving colorectal cancer cell adaptation and invasion in healthy liver tissue. These findings are supported by independent functional and clinical data. Notably, stClinic excels in label annotation through zero-shot learning and facilitates multi-omics integration by relying on other tools for latent feature initialization.

The interrelated, coexisting, and competitive interactions between different cell types within the tumor microenvironment (TME) or cellular niches drive cancer cell heterogeneity, clonal evolution, disease progression, and metastasis[1]. These complex interactions—such as those between tumor-associated macrophages (TAMs), cancer-associated fibroblasts (CAFs), and immune cells—form a dynamic network that significantly influences clinical outcomes, including tumor stage, grade, prognosis, and treatment response[2–4]. Studying

[1]School of Life Sciences, Sun Yat-sen University, Guangzhou, China. [2]Institute of Artificial Intelligence, Donghua University, Shanghai, China. [3]Shanghai Collaborative Innovation Center of Endoscopy, Fudan University, Shanghai, China. [4]Key Laboratory of Symbolic Computation and Knowledge Engineering of Ministry of Education, Jilin University, Changchun, China. [5]System Biology Lab for Metabolic Reprogramming, Department of Human Genetics and Cell Biology, School of Medicine, Southern University of Science and Technology, Shenzhen, China. [6]Department of Pathology, Changzheng Hospital, Secondary Military Medical University, Shanghai, China. [7]College of Computer Science and Technology, Jilin University, Changchun, China. [8]School of Mathematical Sciences, Shanghai Jiao Tong University, Shanghai, China. [9]School of AI, Shanghai Jiao Tong University, Shanghai, China. [10]Key Laboratory of Systems Biology, Shanghai Institute of Biochemistry and Cell Biology, Center for Excellence in Molecular Cell Science, Chinese Academy of Sciences, Shanghai, China. [11]Key Laboratory of Systems Health Science of Zhejiang Province, School of Life Science, Hangzhou Institute for Advanced Study, University of Chinese Academy of Sciences, Chinese Academy of Sciences, Hangzhou, China. [12]West China Biomedical Big Data Center, Med-X Center for Informatics, West China Hospital, Sichuan University, Chengdu, China. ✉e-mail: zuochm@mail.sysu.edu.cn; lnchen@sjtu.edu.cn

cellular niches, rather than focusing solely on individual cell-states, is crucial for capturing the collective behaviors and emergent properties of the tumor ecosystem, which provide a more comprehensive understanding of cancer dynamics and pave the way for precision medicine[4–6]. Identifying clinically relevant niches allows for targeted interventions in the pro-TME, thereby maximizing therapeutic benefits[7].

Current spatial multi-omics technologies, encompassing transcriptome, genome, epigenome, proteome, and metabolome, retain omics profiles within spatial context, facilitating the exploration of tissue cellular niches and molecular heterogeneity[8,9]. Spatially resolved transcriptomics (SRT) data, particularly within tumors, have become increasingly popular for investigating the role of the TME in disease progression[10–13]. Previously, we employed graph neural network (GNN)-based models to dissect spatiotemporal TME heterogeneity from SRT data, by analyzing its intracellular molecular networks and intercellular cell-cell communication[11,14]. With accumulating SMSMO data accumulated across diverse tumor types[12,15–18], there is an urgent need to systematically unravel cellular niches from diverse patients while predicting their associations with clinical outcomes. However, integrating SMSMO data encounters challenges such as data scale and diversity[19], inter-patient heterogeneity[10], and a limited sample set.

Recently, several computational methods have emerged to analyze SMSMO data. Specifically, (i) integration of SRT data from adjacent slices of a tissue (homogeneous integration): SpaGCN[20] identifies spatial domains across multi-slices with coherent gene expression and histology but lacks batch-effect correction capabilities. PASTE[21,22] aligns or integrates adjacent slices using optimal transport theory, followed by STitch3D[23] and GraphST[24], which construct unified neighbor graphs with three-dimensional (3D) spatial locations inferred by PASTE and apply GNNs for integration and spatial domain identification. However, the linear alignments in PASTE, and its derivatives STitch3D and GraphST, struggle to capture heterogeneity within the TME across diverse slices; (ii) integration of SRT data from slices across diverse tissues (heterogeneous integration): SEDR[25] combines autoencoder and GNN to integrate gene expression and spatial location for spatial domain identification, while PRECAST[26] performs dimension reduction and spatial clustering with straightforward projections. STAligner[27] integrates a graph attention autoencoder and spot triplets to identify shared and specific spatial domains across diverse SRT datasets. Yet, the reliance on spot relations across diverse slices may hinder their accurate identification of niches in heterogeneous patients. BANKY[28] integrates the raw gene expression with two additional gene expression matrices (one based on a weighted mean of expression of neighboring cells and another using an azimuthal Gabor filter), and reduces them to low-dimensional features using PCA. However, the linear approach struggles to capture the complexity inherent in the combined data; (iii) integration of spatial multi-omics data from the same or different slices: CellCharter[19] and SLAT[29] preprocess multi-omics data using scVI[30] and GLUE[31], followed by graph modeling to learn shared features; and (iv) integration with clinical data: CytoCommunity[32] integrates spatial location and cell phenotype with a GNN to identify condition-specific spatial domains. However, it is primarily designed for single-cell spatial protein data, and is not suitable for high-dimensional spatial omics data (e.g., transcriptome or epigenome) due to limited cell-type annotation and sample size[33]. More importantly, none of these methods have attempted to explain how the TME or niche influences clinical outcomes. Therefore, there is a lack of computational methods capable of integrating both SMSMO and clinical data to identify niches across diverse tissues while predicting clinically related niches.

In this work, we propose stClinic, a dynamic graph learning model that integrates SMSMO and phenotype data to analyze niches in diverse populations. It (i) identifies shared and condition-specific niches, (ii) evaluates their significance in phenotype prediction, (iii) transfers labels from a reference set using zero-shot learning, and (iv) integrates multi-omics data from the same or different slices. To accurately leverage inter-spot relations within and across slices, stClinic aggregates messages from evolving neighboring nodes with similar feature profiles, facilitating effective learning of batch-corrected features. To overcome sample size limitations, stClinic directly relates niche to clinical outcomes by representing each slice using a niche vector characterized by six geometric statistical measures relative to the population. Crucially, a pre-trained encoder can map new samples into a common feature space as the reference set without fine-tuning, enabling label transfer. Additionally, stClinic's flexible input capabilities allow it to leverage latent features from other single-cell multi-omics tools (e.g., MultiVI[34] and Seurat[35]) into dynamic graphs, thereby enhancing spatial multi-omics alignment and fusion. stClinic demonstrates its versatile applications by detecting shared and clinically related niches from 96 tissue slices, including breast cancer, colorectal cancer (CRC), and liver metastasis (LM). stClinic reveals distinct niches such as TAMs in aggressive tumors, B and plasma cells in favorable prognoses, and *SPP1*+ *MTRNR2L12*+ myeloid cells and CAFs in CRC adaptation and invasion. These findings are supported by independent functional and clinical data. Importantly, such unsupervised and supervised models in stClinic provide a flexible framework for decoding niches by integrating spatial epigenomics, proteomics, and mass spectrometry imaging data.

## Results

### Overview of stClinic

stClinic dissects cellular niches in diverse populations by integrating SMSMO and clinical data through five key components (Fig. 1a–d): (i) learning batch-corrected features from multi-slice data using dynamic graphs; (ii) assessing niche importance in phenotype prediction via attention-based supervised learning; (iii) transferring labels from reference samples by zero-shot learning; (iv) extracting joint features of multi-omics data from the same slice; and (v) learning aligned features of multi-omics data from different slices.

In the unsupervised learning task (Fig. 1b, d), stClinic models omics profiling data ($X$) from multi-slices as a joint distribution $p(X, A, z, c)$, where $c$ represents one of the components within a Gaussian Mixture Model (GMM) comprising $K$ clusters (see "Selecting the number of clusters in GMM"), and $z$ stands for batch-corrected features characterizing biological variations among spots across multi-slices. stClinic employs a variational graph attention encoder (VGAE) to transform $X$ and an adjacency matrix ($A$) (i.e., a unified graph) into $z$ on the Mixture-of-Gaussian (MOG) manifold. The adjacency matrix is constructed by incorporating spatial nearest neighbors within each slice and feature-similar neighbors across slices (Supplementary Fig. 1). It then maps $z$ through $L$ one-layer slice-specific decoders[36,37] to both omics profiling data and the unified graph. To mitigate the influence of potential false neighbors (i.e., dissimilar profiles) in learning embeddings, stClinic iteratively removes links between spots from different GMM components.

To link clusters/niches with clinical outcomes in a limited sample population (Fig. 1c, d), stClinic represents each slice with a niche vector using attention-based statistical measures. For each cluster in each slice, six metrics are calculated to capture variations: mean, variance, maximum, and minimum of the UMAP[38] embeddings in the combined feature space $z$, and proportions within and across all other slices. The weights of each cluster in phenotype prediction can then be determined by the parameters of the linear layer.

Once trained on a reference set, stClinic's frozen graph encoder can seamlessly map tissue samples into the same embedding space, facilitating label transfer from the reference set through zero-shot learning (Fig. 1d). Additionally, with its adaptable input features,

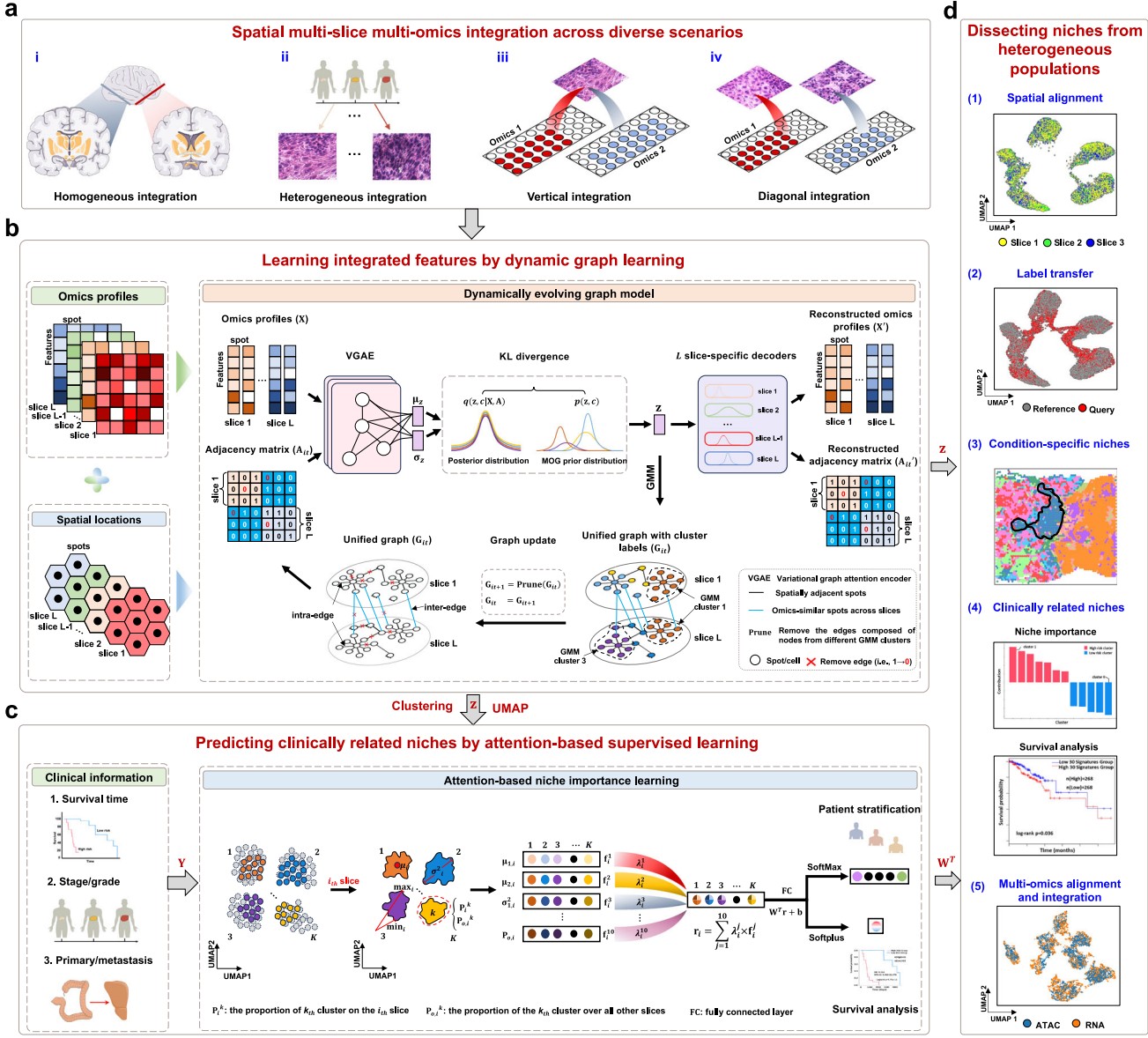

**Fig. 1 | Overview of stClinic. a** stClinic integrates multi-slice omics data from the same tissue or different tissues, as well as multi-omics data from the same slice or different slices/technologies. Vertical integration aligns and integrates multi-omics data within the same slice, while diagonal integration does so across different slices. **b** Given multi-slice omics profiles (**X**) and spatial location (**S**) data as input, stClinic learns batch-corrected latent features (**z**) using a dynamically evolving graph, guided by a Mixture-of-Gaussians prior through Kullback-Leibler (KL) divergence regularization. **c** Given **z** and clinical data (**Y**) as input, stClinic quantifies the weights (**W**$^T$) of each cluster in clinical outcome prediction by representing each slice using a niche vector characterized by six geometric statistical measures relative to the population. **d** Integrated features **z** and weights (**W**$^T$) serve various purposes in dissecting niches from heterogenous tissues: identifying shared and condition-specific niches, assessing niche importance in phenotype prediction, transferring labels from the reference through zero-shot learning, and annotating labels across different types of omics datasets.

stClinic relies on latent features extracted from multi-omics data using prior tools like MultiVI[34] and Seurat[35], facilitating label annotations across various omics datasets (Fig. 1d).

## stClinic enables accurate alignment of SRT datasets across diverse samples

To thoroughly evaluate the performance of stClinic, we analyzed 12 human dorsolateral prefrontal cortex (DLPFC) slices generated using Visium technology from three donors (Supplementary Table 1)[39]. Each slice was annotated with six (or four) layers and white matter (WM) from a previous study, serving as the ground truth for evaluating clustering accuracy (Fig. 2a)[39]. To emphasize the advantages of batch-corrected features learned from dynamic graphs, we also evaluated

features learned from a fixed graph (stClinic_fix). We compared stClinic with five recent methods (SEDR, GraphST, STitch3D, PRECAST, and STAligner). We used the respective algorithms for each method to predict clusters and applied the mclust algorithm[40] to stClinic_fix and stClinic (Supplementary Table 2). The adjusted rand index (ARI)[41] and normalized mutual information (NMI)[42] measure cluster consistency with the ground truth, while the average silhouette width (ASW) evaluates cluster separation. The F1 score[26], derived from ASW, assesses both cluster separation and slice mixing. The cell-type local inverse Simpson' Index (cLISI)[43] and integration LISI (iLISI)[26] quantify the mixing of clusters and slices in the neighborhood, respectively. Spot embeddings were visualized by projecting them into two UMAP spaces.

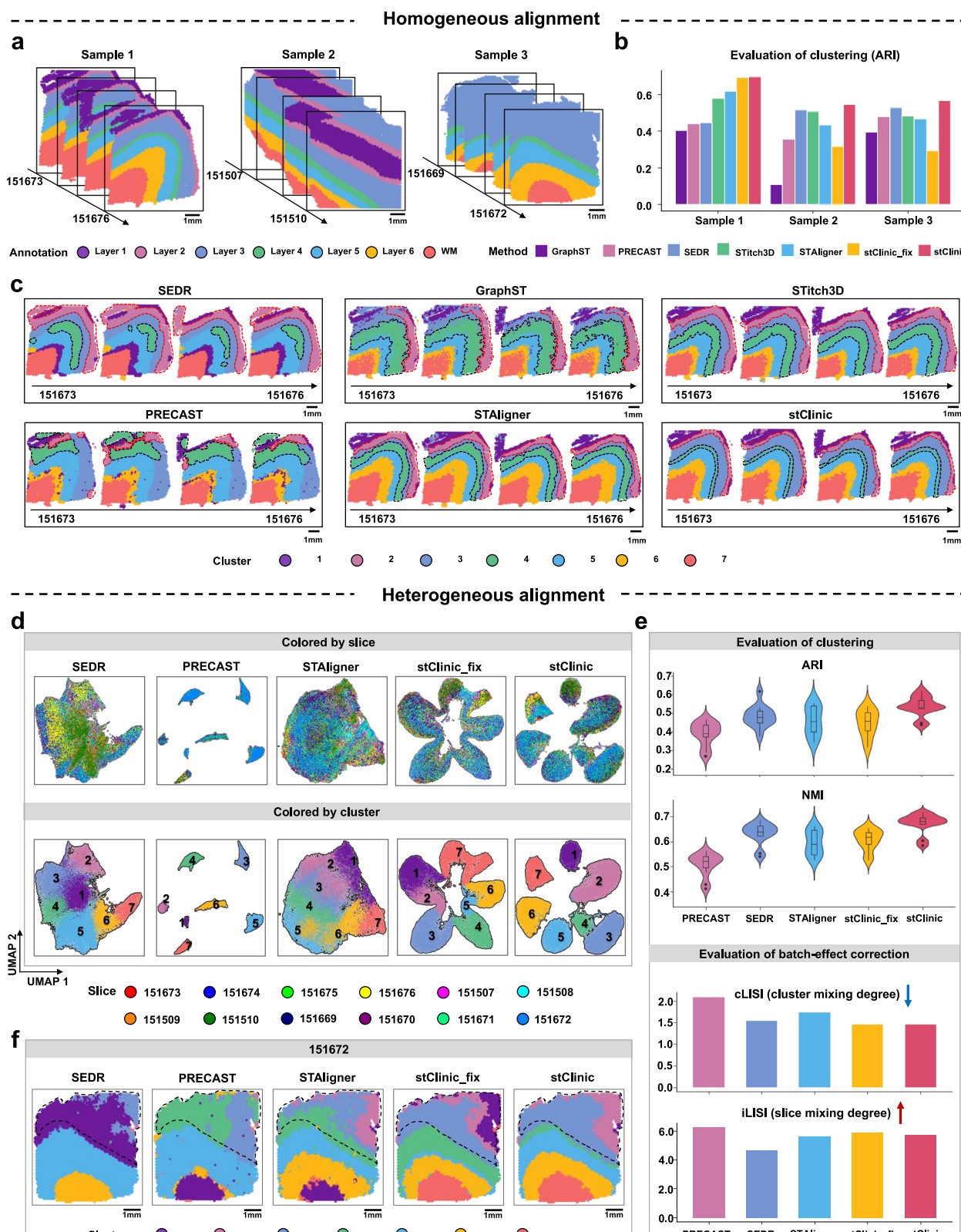

We corrected batch effects of four adjacent slices for each sample. In summary, we found that (i) PRECAST achieves higher ASW and F1 scores than stClinic, indicating better cluster separation. However, stClinic excels in ARI and NMI scores, reflecting greater clustering accuracy. For example, in Sample 1, stClinic accurately distinguishes between Layers 4 and 5, as well as Layers 1 and 2 (Fig. 2b, c and Supplementary Figs. 2–4); (ii) compared to stClinic_fix, latent features

from different slices are more effectively mixed in stClinic, and identical clusters are more closely bound together, demonstrating the effectiveness of aggregating information from dynamic neighboring nodes across slices (Supplementary Figs. 2b, 3a and 4a); and (iii) for cLISI, SEDR and STAligner outperform stClinic in Sample 2, but stClinic surpasses all methods in Samples 1 and 3. stClinic's iLISI score is comparable to those of SEDR, Stitch3D, and STAligner, although all

**Fig. 2 | stClinic is able to align multiple SRT datasets across diverse samples.**
**a** Manual annotation of the 12 slices across three samples on the human DLPFC dataset, including six (or four) layers and WM. **b** Bar plot illustrating clustering accuracy in terms of ARI on three samples by SEDR, GraphST, STitch3D, PRECAST, STAligner, stClinic_fix, and stClinic. **c** Spatial domains detected by SEDR, GraphST, STitch3D, PRECAST, STAligner, and stClinic on the Sample 1. **d** UMAP visualization of the latent features by SEDR, PRECAST, STAligner, stClinic_fix, and stClinic across all 12 slices. In the top and bottom panels, the colors represent the slices and clusters, respectively. **e** Comparison of the seven methods (SEDR, GraphST,

STitch3D, PRECAST, STAligner, stClinic_fix, and stClinic) regarding accuracy of clustering (ARI and NMI) and batch-effect correction (cLISI and iLISI), on $N = 12$ slices. It's noted that lower cLISI values indicate better correction for cell-type mixing, while higher iLISI values indicate better correction for batch mixing. For each boxplot, the center line, box limits, and whiskers separately indicate the median, upper and lower quartiles, and $1.5 \times$ interquartile range. **f** Spatial domains identified on slice 151672 by five methods (SEDR, PRECAST, STAligner, stClinic_fix, and stClinic) under the condition of integrating 12 slices, respectively. Layer 3 is outlined in black. Source data are provided as a Source Data file.

four methods underperform relative to GraphST and PRECAST (Supplementary Fig. 2a).

To further investigate stClinic's ability to integrate heterogeneous tissue slices, we analyzed 12 DLPFC slices from three samples, excluding GraphST and STitch3D as they only integrate slices from the same tissue. We observed that (i) although the clusters of PRECAST are more separated than those of stClinic, stClinic's clusters are more consistent with the ground truth, as evidenced by higher ARI and NMI scores (Fig. 2d, e and Supplementary Fig. 5a); (ii) UMAP embeddings of SEDR, STAligner, and PRECAST show over-corrected and disordered cortical layer structures, whereas stClinic effectively integrates spots from the same layer across 12 slices. This outcome is characterized by exceptional cLISI and iLISI scores, as well as superior histological and transcriptomic similarities compared to other methods (Fig. 2d, e, Supplementary Fig. 5b, c and Supplementary Note 1); (iii) stClinic outperforms stClinic_fix in aligning identical clusters from heterogeneous slices, resulting in a tighter and more homogeneous mix (Fig. 2d and Supplementary Fig. 5a–c); and (iv) for sample 3 annotated with Layers 3–6 and WM, stClinic identifies more accurate and smoother laminar patterns than other methods, and can even detect Layers 2 and 1 within the annotated Layer 3. This finding is consistent with the distribution of Layer 1- and 2-marker genes such as *AQP4* and *HPCAL1*[39] (Fig. 2f and Supplementary Fig. 5d).

Moreover, an integrative analysis of seven consecutive sections of the 3D hippocampal structure profiled by Slide-seq[44] demonstrated that, although stClinic does not achieve the highest ASW, F1 score, or iLISI, it consistently outperforms other methods in capturing cluster patterns, as supported by a more accurate gene expression distribution within clusters (Supplementary Fig. 6a–e).

Overall, stClinic effectively aligns identical clusters across slices in the low-dimensional feature space, enabling the dissection of heterogeneous cellular niches across different tissues.

## stClinic uncovers intra-tumor niches missed by competing methods

To demonstrate stClinic's ability to discern heterogeneous niches within complex disease tissues, we applied it to analyze two human Luminal B breast cancer slices, BAS1 and BAS2, as published by 10X Genomics (Fig. 3a and Supplementary Table 1). These samples were annotated into 20 groups, categorized into four histological types: invasive carcinoma (IDC), carcinoma in situ (CIS), tumor edge, and healthy tissue. We benchmarked stClinic against SpaGCN, Seurat, SLAT, SEDR, STAligner, PRECAST, and BANKSY. The low-dimensional features generated by stClinic, Seurat, SpaGCN, and STAligner were used to identify spatial domains using the Louvain algorithm, while the other three methods employed their default clustering methods (Supplementary Table 2). The results were visualized in two-dimensional UMAP spaces.

By comparison, we found that (i) while PRECAST achieves a higher F1 score, stClinic demonstrates superior ARI performance, exceeding PRECAST by more than 0.1. This highlights stClinic's ability to capture local details and maintain pairwise annotation consistency. Notably, stClinic uniquely detects tumor boundaries, such as cluster 17 (indicated by black arrows), which is characterized by over-expression of oxidative phosphorylation and glycolysis genes, potentially reflecting

tumor cell invasion and metastasis[45] (Fig. 3a–e and Supplementary Fig. 7a); (ii) BANKSY achieves a slightly higher NMI score than stClinic, reflecting its strength in capturing global categorical relationships. However, BANKSY fails to identify local small domains across slices, such as IDC_1 (outlined in black) and CIS_2 (outlined in red) (Fig. 3a, c); (iii) for cLISI, BANKSY, SLAT, and SEDR outperform stClinic, while for iLISI, PRECAST and Seurat perform better than stClinic (Fig. 3c); (iv) SpaGCN, despite its ability to identify spatial domains, struggles to align several common niches across slices due to its lack of batch-effect correction capabilities (Fig. 3a, b); and (v) stClinic demonstrates superior robustness to variations in the number of clusters, consistently maintaining overall high clustering and batch-effect correction performance in terms of F1 score, ARI, and NMI (Supplementary Fig. 7b).

In summary, stClinic effectively integrates multi-slice SRT data within a batch-corrected feature space, enabling accurate and reliable analysis of tumor heterogeneity.

## stClinic evaluates niche malignancy through prognosis analysis

We demonstrated the ability of stClinic to assess malignancy levels in diverse niches using survival time, by analyzing 43 slices from 23 triple-negative breast cancer (TNBC) patients[46] (Supplementary Table 1). These slices, marked by regions of tumor, fibrosis, necrosis, adipose tissue, and immune infiltrate, were stratified into low-risk and high-risk groups based on the median overall survival time. The log-rank test revealed a significant survival difference between the groups (Fig. 4a, $p = 1.66e\text{-}08$), indicating that the data can effectively distinguish survival-associated clusters. We compared stClinic with SEDR, PRECAST, and STAligner to identify clusters from these 43 slices (Supplementary Table 2), and subsequently assessed the importance of different clusters in predicting survival time.

In short, we observed that (i) UMAP embeddings of SEDR, STAligner, and PRECAST are structurally unordered, whereas stClinic achieves a uniform mixture of spots across 43 slices while maintaining distinct cluster separation (Fig. 4b and Supplementary Fig. 8a); (ii) stClinic's clusters closely align with pathological annotations: clusters 1, 2, 4, 9, and 10 reside in in the tumor region, clusters 6, 7, 8, 11, and 12 in lymphoid cell and fibrosis regions, and clusters 3 and 5 in fibrosis and myeloid cell regions (Fig. 4c–e and Supplementary Figs. 8b and 9–13); (iii) stClinic and STAligner identify detailed structures within the dense lymphoid infiltrate and fibrosis regions using sub-clustering analysis, while SEDR performs less effectively, as shown in slice 15. Both stClinic and STAligner detect two clusters enriched in T and B cells or myeloid cells. Notably, stClinic exhibits significant PVL enrichment in one cluster compared to the other two, whereas STAligner displays PVL enrichment in one cluster relative to a single other cluster (Supplementary Fig. 14a–c); (iv) stClinic accurately predicts survival time, achieving a median of concordance index (C-Index) of 0.855 by using seven-fold cross-validation, with a significant difference between high-risk and low-risk groups (hazard ratio (HR): 9.354, 95% confidence interval (CI): 3.963–22.076, log-rank test, $p = 4.75e\text{-}12$) (Fig. 4f and Supplementary Fig. 15); and (v) cluster 10 exhibits the highest positive weight for survival prediction, while cluster 6 has the lowest negative weight. Notably, we found that positive-weight clusters show enrichment of cancer and macrophage cells, associated with poor

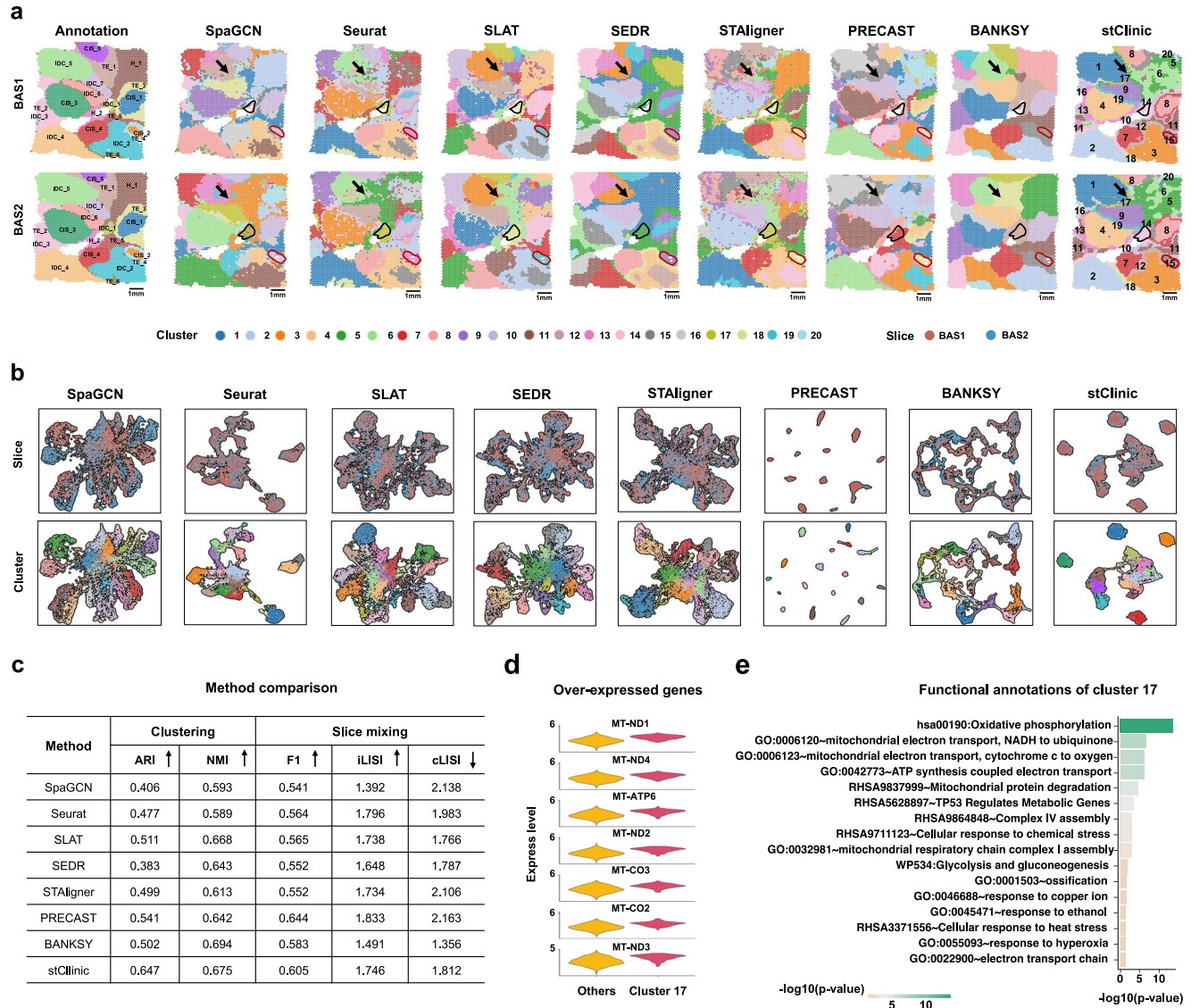

**Fig. 3 | stClinic enables the identification of intra-tumoral niches in two human Luminal B breast cancer slices: BAS1 and BAS2. a** Manual pathological annotation of Hematoxylin and Eosin (H&E)-stained plots and clusters identified by SpaGCN, Seurat, SLAT, SEDR, STAligner, PRECAST, BANKSY, and stClinic. **b** UMAP visualization of latent features for the two slices generated by eight methods. Colors in the top and bottom panels represent slices and clusters, respectively. **c** Comparison of clustering accuracy (ARI and NMI) based on manual annotations, slice mixing accuracy using annotations (cLISI) and slice numbers (iLISI), and combined cluster separation and slice mixing performance (F1 score) across eight methods. **d** Over-expressed genes in cluster 17 compared to other clusters. **e** Functional annotation of over-expressed genes in cluster 17 relative to other clusters. Unadjusted one-sided Fisher's exact test. Source data are provided as a Source Data file.

prognosis[47], while negative-weight clusters are enriched with plasma and B cells, linked to improved prognosis[48,49] (Fig. 4c, g). Collectively, these findings suggest that clusters 10 and 6 may represent high-risk and low-risk niches, respectively.

To support our findings, we adopted independent approaches: (1) cluster 10 cells exhibit over-expression of genes related to tumor metastasis and epithelial-mesenchymal transition[50–52], such as *TROP2*, *TM4SF1*, and *FABP5*. These genes are involved in functions linked to tumor malignancy, including cell division, cell cycle, cell migration, hypoxia, cell proliferation, mechanical stimulus, and degradation of the extracellular matrix. Moreover, the mean expression level of 30 signature genes is significantly associated with shorter survival in breast cancer, as demonstrated by data from the TCGA database (log-rank test, $p = 0.036$) (Fig. 4h–j); (2) conversely, cluster 6 cells over-express IgG genes linked to improved prognosis[53], along with tumor-suppressive genes like *TIMP3* and *LYZ* [54–56]. This cluster mainly comprises CD8+ T cells, memory B cells, and dendritic cells (DCs),

suggesting its crucial role in the tumor immune response through mechanisms such as CD8+ T-mediated cytotoxicity, antigen presentation by DCs, and antibody production by B cells. Additionally, the mean expression level of 30 signature genes is significantly associated with longer survival in breast cancer from the TCGA database (log-rank test, $p = 0.032$), suggesting that the immune-active profile of this cluster may serve as a prognostic biomarker for better outcomes in patients with stronger anti-tumor immune responses (Fig. 4c, h, i and Supplementary Figs. 16 and 17a); and (3) cluster 10 cells exhibit higher CNV levels compared to cluster 6 cells (Fig. 4k and Supplementary Fig. 17b–d). These results indicate that clusters 10 and 6 are associated with unfavorable and good prognosis, respectively.

Overall, stClinic offers a valuable methodological approach to assess niche malignancy by predicting survival time, thereby enhancing prognostic assessment and potentially uncovering targets for clinical immune therapy. These hold promise for improving patient outcomes in cancer treatment.

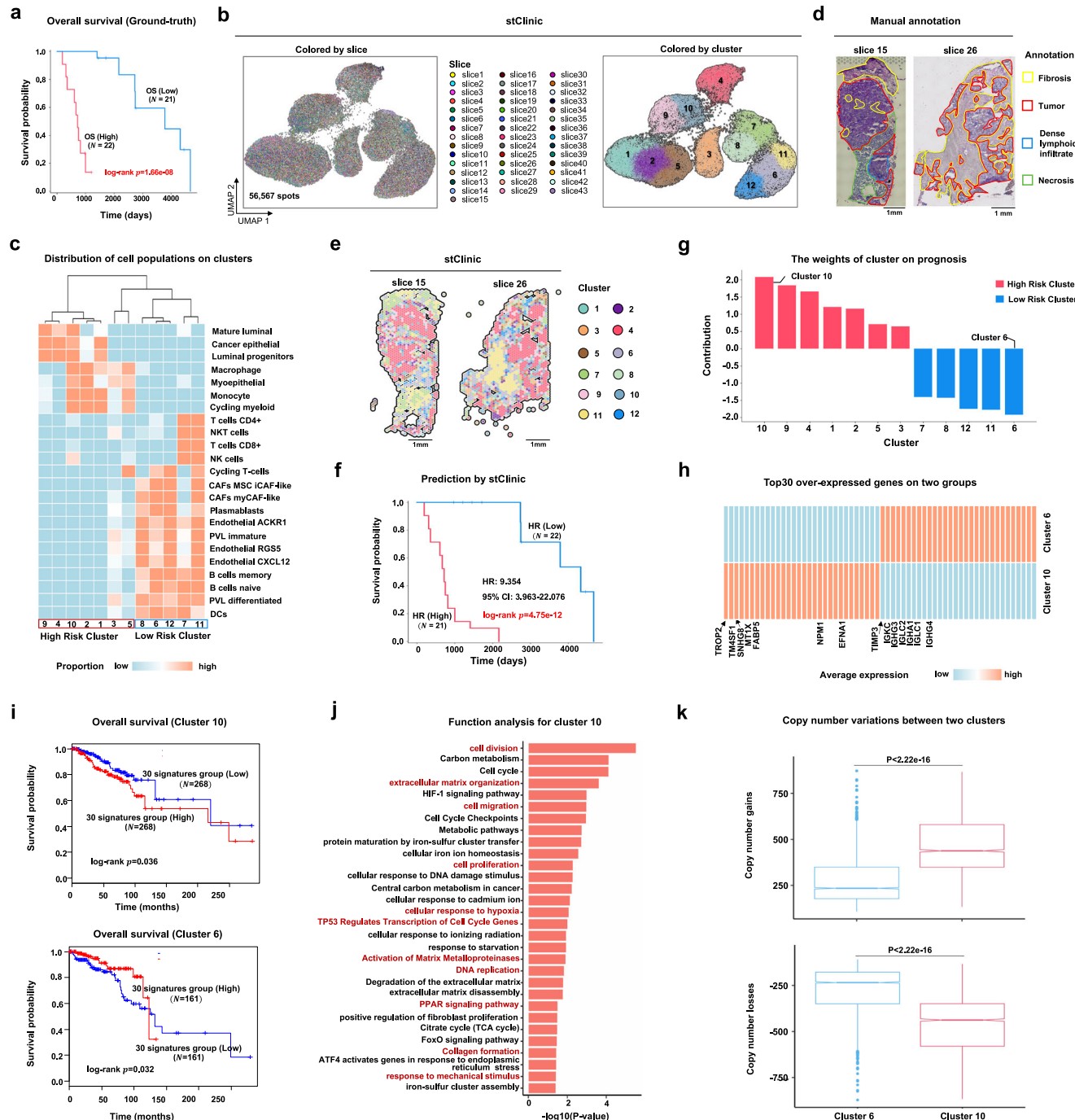

**Fig. 4 | stClinic evaluates the malignancy level of niches on the 43 TNBC slices.**
**a** Kaplan–Meier survival curve for 43 breast cancer slices, with slices stratified into low-risk and high-risk groups based on their median overall survival time. Unadjusted two-sided log-rank test. **b** UMAP visualization of the latent features by stClinic on 43 slices, where the colors in the left and right panels indicate slices and clusters, respectively. **c** Heatmap showing the cell-type proportions on spatial domains by stClinic. **d** H&E plot of slices 15 and 26. Slice 15 was annotated with fibrosis, necrosis, tumor, and dense lymphoid infiltration regions, while slice 26 was annotated with fibrosis and tumor regions. **e** Spatial domains identified by stClinic on slices 15 and 26. **f** Kaplan–Meier survival curves for 43 breast cancer slices, with slices classified into low-risk and high-risk groups based on their median hazard ratio predicted by stClinic. Unadjusted two-sided log-rank test. **g** Bar plot

displaying the weights of cluster in prognosis by stClinic. **h** Heatmap depicting the average gene expression of the top 30 over-expressed genes for clusters 10 and 6. **i** Overall survival rate of patients with the low or high expression patterns of the top 30 over-expressed genes for clusters 10 (top panel) and 6 (bottom panel) in breast cancer data from TCGA by GEPIA2[89]. Unadjusted two-sided log-rank test. **j** Functional enrichment analysis of the over-expressed genes in cluster 10. Unadjusted one-sided Fisher's exact test. **k** Boxplot showing the total copy number gains and losses per spot in clusters 10 ($N = 2717$) and 6 ($N = 5124$), as inferred by CopyKAT[90]. Unadjusted one-sided Wilcox rank-sum test. 2.22e-16 represents a very small number, effectively close to zero. For each boxplot, the center line, box limits, and whiskers separately indicate the median, upper and lower quartiles, and 1.5 × interquartile range. Source data are provided as a Source Data file.

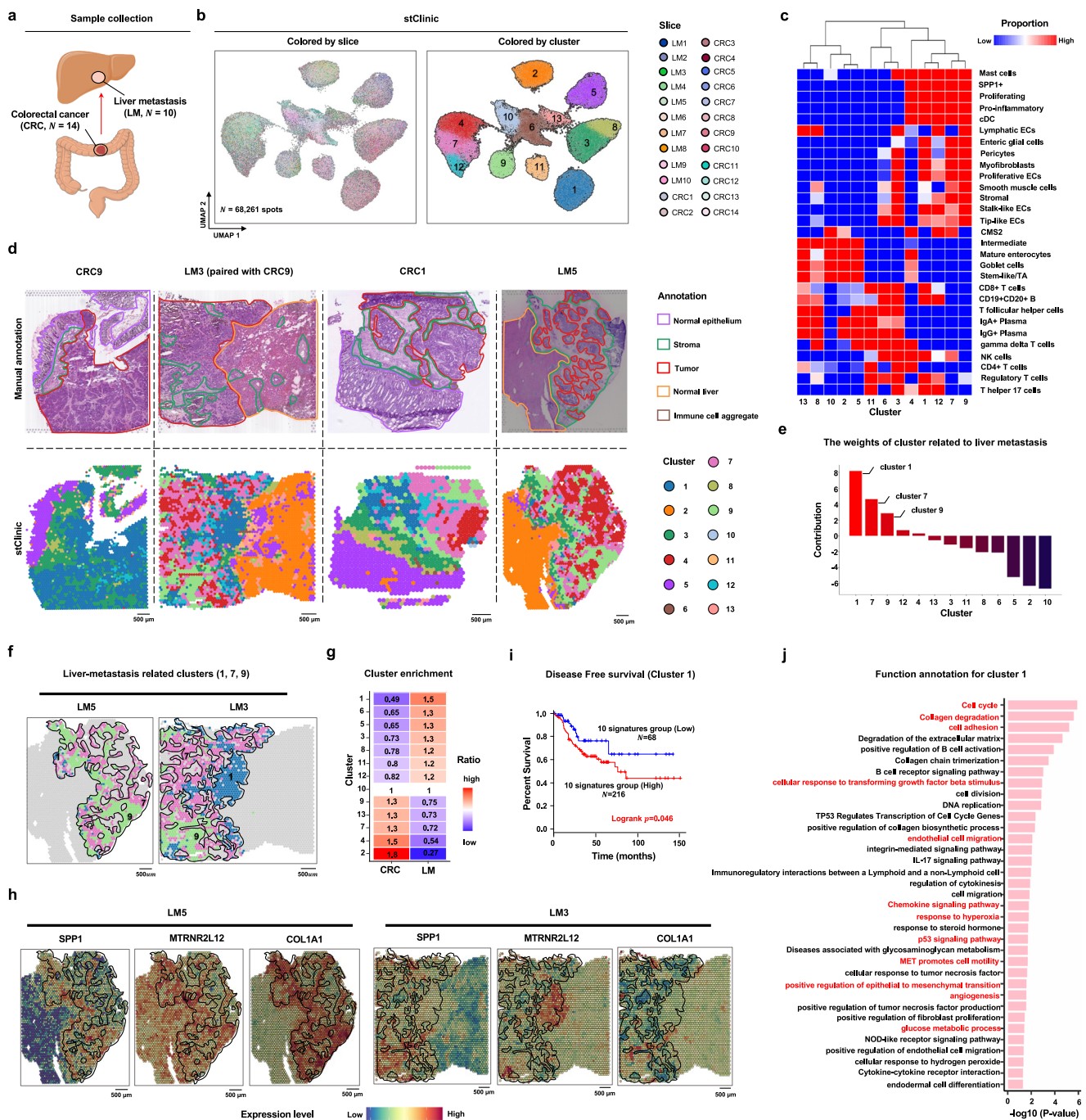

**Fig. 5 | stClinic enables the identification of metastasis-relevant TMEs by integrating primary colorectal cancer and liver metastasis datasets. a** Sample set of 14 primary CRCs and 10 LM slices. **b** UMAP visualization of the latent features by stClinic on 24 slices, where the colors in the left and right indicate slices and clusters, respectively. **c** Heatmap showing the cell-type proportions on spatial domains identified by stClinic. **d** H&E plot of four representative slices: CRC1, CRC9, LM3, and LM5, where CRC9 and LM3 are from the same patient. The top panel shows the pathological annotations of four slices, while the bottom panel indicates the spatial domains identified by stClinic. **e** Bar plot showing the weights of different clusters in liver metastasis by stClinic. **f** Spatial distribution of liver metastasis-related clusters (1, 7, 9) on LM5 and LM3. **g** Heatmap of cluster enrichment on CRC and LM slices identified by stClinic. **h** Spatial expression of representative genes (*SPP1*, *MTRNR2L12*, and *COL1A1*) for metastasis-related cells, for slices LM5 and LM3 data denoised by stClinic. **i** Disease-free survival rate of patients with high and low expression patterns of the top 10 over-expressed genes for cluster 1 in colon cancer data from TCGA by GEPIA2[89]. Unadjusted two-sided log-rank test. **j** Functional enrichment analysis of the over-expressed genes in cluster 1. Unadjusted one-sided Fisher's exact test. Source data are provided as a Source Data file.

## stClinic identifies metastasis-relevant niches by integrating primary colorectal cancer and liver metastasis data

To validate that stClinic can predict metastasis-related niches, we manually collected 24 slices comprising 14 primary CRCs and 10 LMs[15,57–60], including pairs from the same patient (CRC9/LM3 and CRC10/LM4) (Fig. 5a and Supplementary Table 1). These slices were annotated into five regions: normal epithelium, stroma, tumor, normal liver, and immune cell aggregate[15,57–60]. By applying stClinic to integrate these 24 slices, we identified 13 clusters and assessed the niche importance in distinguishing LM and CRC.

Upon analysis, we found that (i) stClinic's spatial domains align more closely with pathological annotations compared to SEDR, PRE-CAST, and STAligner. For example, clusters 1, 4, 7, 9, and 12 are enriched in the tumor region, cluster 2 in the normal liver region, cluster 6 in the immune cell aggregate region, cluster 5 in the normal epithelium region (Fig. 5b–d and Supplementary Figs. 18–24); (ii) spots belonging to the same clusters across 24 slices are more evenly mixed in stClinic compared to other methods (Fig. 5b and Supplementary Fig. 20a); and (iii) stClinic accurately predicts primary and metastatic cancer categories by niche, achieving an average area under the receiver operating characteristic curve (AUCROC) of 0.914 and an accuracy (ACC) of 0.917, across 24 sets of leave-one-out cross-validation (Supplementary Fig. 25). Notably, cluster 1 exhibits the highest positive weight for classification, suggesting its association with LM from primary CRC (Fig. 5e, f).

To validate our findings, we adopted the following ways: (1) cluster 1 is notably abundant in LM, particularly in the normal liver region (Fig. 5g); (2) cluster 1, enriched with *SPP1+* myeloid and CAF cells, forming an immunosuppressive niche supporting CRC in the liver[61]. Notably, *SPP1+* myeloid cells over-express *MTRNR2L12*, enhancing macrophage survival within the TME, particularly in metastasis[62] (Fig. 5h); (3) the mean expression level of 10-gene signature in cluster 1 is significantly associated with shorter disease-free survival (DFS) in the CRC samples from the TCGA database (log-rank test, $p = 0.046$), suggesting that its potential as a biomarker for predicting disease recurrence or death (Fig. 5i); (4) a random forest model trained on a 10-gene signature successfully classified CRC (stages I–III vs. stage IV) using 90% of the TCGA data, achieving an accuracy of 0.87 and an AUC of 0.74 on the test data. The AUC was significantly higher than those from 1000 random 10-gene sets drawn from ~60,000 genes, indicating non-random predictive value (unadjusted one-sided Permutation test, $p = 0.049$); and (5) cluster 1 shows various functions, including cell cycle, collagen degradation, endothelial cell migration, p53 signaling pathway, angiogenesis, cytokinesis, glucose metabolic process, and hypoxia (Fig. 5j). These results suggest that cluster 1 may suggest a potential role in early normal liver tissue infiltration, this remains a hypothesis that requires further genomic data for confirmation.

In short, stClinic streamlines the prediction of metastasis-related niches by integrating primary CRC and LM data. This paves the way for a deeper understanding of how cancer cells navigate and invade new environments by interacting with the surrounding cell populations.

## stClinic annotates labels from the reference using zero-shot transfer learning

A notable feature of our stClinic model is its seamless label transfer from the reference set without the need for retraining (Fig. 6a). We conducted experiments on the DLPFC dataset to illustrate the effectiveness of stClinic. Specifically, we (i) trained a stClinic model using reference slices 151673–151675; (ii) utilized the frozen graph encoder to map slice 151676 from the same tissue and slice 151507 from different tissues into the same embedding space as the reference; and (iii) predicted spot labels based on their nearest neighbors in slices 151673–151675. For benchmarking, we compared stClinic with Seurat and Geneformer[63], a scRNA-seq data foundation model that supports batch-effect correction and zero-shot learning. The results revealed that (i) stClinic classifies ~70% of spots accurately, compared to Seurat and Geneformer's 16–18% accuracy, showing its superior generalizability in both the same and different tissues (Fig. 6b–d); and (ii) in ~80% of layers, over 50% of spots were accurately predicted into their respective classes by stClinic (Supplementary Fig. 26a, b).

A similar experiment was conducted on heterogeneous cancer samples, training stClinic on VIDC and BAS1, and transferring labels to BAS2 (Fig. 6e). By comparison, we observed that (1) stClinic's UMAP embeddings display greater separation compared to Geneformer and Seurat, as indicated by higher F1 scores (Fig. 6f and Supplementary

Fig. 27a). While Seurat exhibits a higher iLISI, indicating greater slice mixing, stClinic's predictions align more closely with annotations, as demonstrated by superior hLISI, the highest ARI and NMI for BAS1, and consistent cell-type compositions across slices within identical cluster (Fig. 6g, Supplementary Fig. 27b, c, and Supplementary Note 2). Additionally, stClinic uniquely identifies cluster 10 (tumor edge) surrounding cluster 1 (tumor core), enriched in macrophages, memory B cells, and CD4+ T cells, with enhanced complement activation and HLA class II expression, reflecting dual roles in tumor growth[64] and anti-tumor immunity (Supplementary Fig. 27d–g); (2) stClinic achieves the highest ARI and NMI scores for transferred labels on BAS2, whereas Seurat performs the worst (Fig. 6g); (3) stClinic demonstrates strong clustering, batch-effect correction, and label transfer performance across varying numbers of clusters, with the highest ARI score for BAS1 at 14 clusters, which was selected for label transfer analysis (Fig. 6g and Supplementary Figs. 27a and 28); and (4) stClinic distinctly identifies cluster 9 surrounding cluster 6 in BAS1 and accurately transfers these labels to BAS2. In both BAS1 and BAS2, cluster 9 exhibits over-expression of basal cell markers (*KRT5*, *KRT14*, and *KRT17*)[65], and *SERPINA3*[66], and genes associated with extracellular matrix remolding, collagen metabolism, cell migration, tissue repair, inflammatory response, and the PI3K-Akt signaling pathway, underscoring its critical role in promoting tumor cell invasion (Fig. 6h–j).

Collectively, stClinic introduces a framework for dissecting query samples using reference data, facilitating knowledge transfer from prior studies and advancing our understanding of complex biological systems.

## stClinic improves the results for detecting finer structures by integrating multi-omics from the same slice or different slices

Benefiting from stClinic's flexible framework, we explored its capacity to integrate spatial multi-omics data from both the same[67] and different slices[68,69] (Fig. 7a, b). We used MultiVI[34] to map both RNA-seq and ATAC-seq from the P22 mouse brain coronal section[67] into latent features (Supplementary Table 1). For comparison, we separately employed scVI[30] and peakVI[70] to map RNA-seq and ATAC-seq data into latent features. stClinic then extracts low-dimensional features from these profiles and spatial location data. Spatial clusters were subsequently predicted, with CellCharter[19] used for comparison. Our analysis revealed that (i) multi-omics integration tools like stClinic, CellCharter, and MultiVI better approximate true tissue structure compared to single-omics model (Fig. 7c and Supplementary Fig. 29a, b); (ii) stClinic's domains closely align with anatomical structures, such as VL (cluster 12) and islm (cluster 13), consistent with distribution of their corresponding markers: *Dlx1* and *Drd3*, indicating the efficiency in dynamically aggregate information from neighboring nodes within complex tissues (Fig. 7d and Supplementary Fig. 29c–e).

We then investigated whether or not stClinic could effectively align multi-omics data from different slices. For the RNA-seq (from Stereo-seq)[69] and ATAC-seq (from spatial ATAC-seq)[68] on different mouse embryo slices (Supplementary Table 1), we applied Seurat[35] to map them into a shared feature space, generating a feature profile for each slice. Then, we treated this as a multi-slice integration task (Fig. 7b). We also employed GLUE[31], SLAT[29], and MaxFuse[71] for analysis. Comparatively, stClinic demonstrates superior alignment between RNA-seq and ATAC-seq compared to GLUE, SLAT, Seurat, and Max-Fuse, despite the different resolutions between the two technologies, i.e., 0.2 μm for Stereo-seq and 20 μm for spatial ATAC-seq. Furthermore, transferred ATAC-seq labels from RNA-seq data in stClinic exhibited greater consistency with known marker genes, such as *Cntnap5b* for spinal cord, *Dnm3os* for connective tissue, and *Tnn2* for heart (Fig. 7e–g and Supplementary Fig. 29f).

Overall, these findings highlight stClinic's effectiveness in intelligently integrating information from feature-similar neighboring nodes, whether these omics are from the same slice or different slices.

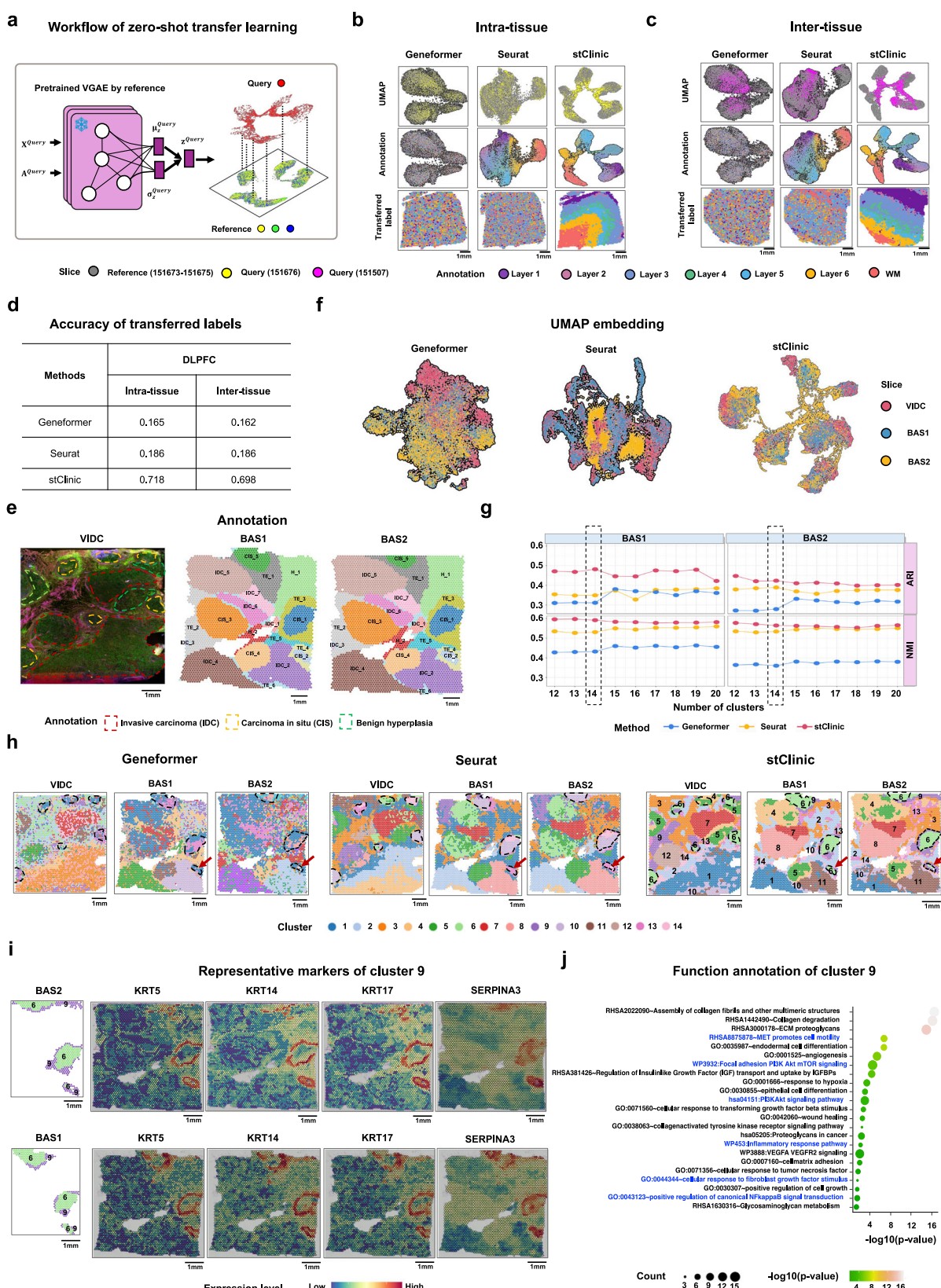

## Discussion

This study presents stClinic, a dynamic graph model for analyzing niches in heterogeneous populations using SMSMO and clinical data, by identifying shared and patient-specific niches, assessing niche significance in clinical outcomes, transferring labels from the reference using zero-shot learning, and integrating multi-omics data from both the same and different slices. stClinic aggregates information from evolving neighboring nodes with similar profiles, enabling the learning of batch-corrected and biologically coherent representations. It introduces six geometric statistical measures to quantify cluster/niche patterns—presence, proportion, and distribution—using UMAP embeddings of batch-corrected features. UMAP's preservation of local

**Fig. 6 | stClinic facilitates label transfer from a reference set using zero-shot learning. a** A pre-trained encoder maps new samples into the same feature space as the reference set without fine-tuning, enabling label transfer. **b** UMAP visualization of latent features by Geneformer, Seurat, and stClinic across four slices (151673–151676). Slices 151673–151675 serve as the reference set for training stClinic and Seurat models, while slice 151676 is treated as the query slice. The top panel colors represent slices, the middle panel colors indicate spot annotations, and the bottom panel shows the spatial distribution of predicted labels for slice 151676. **c** UMAP visualization of latent features by Geneformer, Seurat, and stClinic across four slices (151673–151675 and 151507), using the same models as in (**a**). Slice 151507 is treated as the query sample. The top panel colors represent slices, the middle panel colors indicate spot annotations, and the bottom panel shows the spatial distribution of predicted labels for slice 151507. **d** Table showing the accuracy of

three methods for intra- and inter-tissue label transfer in DLPFC samples. **e** Immunofluorescence plot and annotations of VIDC, along with detailed histological annotations of BAS1 and BAS2. **f** UMAP visualization of latent features by Geneformer, Seurat, and stClinic across VIDC, BAS1, and BAS2, with VIDC and BAS1 as the reference set and BAS2 as the query slice. Noted that colors indicate slices. **g** Line plot displaying ARI and NMI scores for clusters predicted by Geneformer, Seurat, and stClinic compared to annotations across varying numbers of clusters in BAS1 and BAS2. **h** Spatial distributions of clusters predicted or transferred by three methods in VIDC, BAS1, and BAS2. **i** Spatial distribution of cluster 9 and its marker genes (*KRT5*, *KRT14*, *KRT17*, and *SERPINA3*) in BAS1 and BAS2. **j** Functional annotation of over-expressed genes in cluster 9. Unadjusted one-sided Fisher's exact test. Source data are provided as a Source Data file.

and global structures ensures consistent cluster alignment across slices, enabling predictive statistics that link niche features to clinical outcomes. Notably, stClinic directly connects spatial niches to clinical outcomes through multi-slice omics integration. Its shared encoder seamlessly maps new samples into the common feature space of the reference set for label transfer through zero-shot learning. Through versatile input features, stClinic incorporates latent features from other multi-omics tools, enabling label annotations across diverse datasets.

Benchmark comparisons between stClinic and the other six methods on the human DLPFC dataset demonstrated that stClinic's latent features of the same clusters from different slices are more effectively integrated, facilitating the dissection of heterogeneous cellular niches. Further evaluations on human breast, colorectal, and LM cancer samples highlighted the unique benefits of stClinic in assessing the niche importance for phenotypic prediction. stClinic evaluates niche malignancy through prognosis analysis, identifying high-risk niches marked by TAMs linked to cell proliferation and migration, and low-risk niches enriched with B and plasma cells, signifying immune cell activation. Additionally, stClinic predicts a niche promoting CRC cells adaptive to normal liver tissue, featuring *SPP1*+ *MTRNR2L12*+ myeloid cells and CAFs by integrating primary and metastasis cancers. These findings provide valuable insights for uncovering targets in clinical immune therapy and improving prognostic assessment, ultimately leading to enhanced patient outcomes.

In this study, we focused on analyzing spot-level SRT data generated using Visium and also demonstrated the effectiveness of stClinic in revealing tissue structures across multiple technologies. By analyzing two mouse embryo slices—seqFISH (E8.75)[72] and Stereo-seq (E9.5)[69]—stClinic successfully identified the Otocyst at both stages, marked by key genes such as *Gbx2*, *Dll3*, *Lfng*, and *Fst*[73]. Notably, the Otocyst was not detected by other methods or prior analyses[69,72] (Supplementary Fig. 30a–g). Additionally, we have shown the versatility of stClinic by analyzing two breast cancer slices, profiled by 10X Xenium and Visium, from the same tissue[74]. The results highlighted the efficiency of stClinic in aligning identical niches between the slices, and also identifying a greater diversity of cancer cell-states. For example, it detected four clusters within the ductal CIS regions (DCIS #1 and DCIS #2) in the Xenium slice, whereas other methods detected only two or three clusters. These findings were further validated by differential gene expression analysis and cell-type enrichment (Supplementary Fig. 31a–g). With the expansion of spatial multi-omics technologies, such as spatial ATAC-seq[75], MSI[76], and transcriptome-protein[77], or transcriptome-chromatin accessibility profiling[67,78], stClinic has shown robustness and adaptability for integrative analyses across datasets of varying resolutions and scales.

We benchmarked the running time of stClinic on the simulated datasets by subsampling spots from the CRC and LM datasets. Compared to STAligner, stClinic is faster, taking only 19 min to integrate a dataset of 68K spots from 24 slices. This highlights the efficiency of

stClinic in analyzing large-scale SRT datasets from multi-slices (Supplementary Fig. 32).

There are still some limitations in stClinic. Specifically, (1) the removal of links between spots from distinct GMM components enhances the separation of latent features across clusters. However, exploring spatiotemporal relationships between clusters is essential for quantifying biological systems and predicting their complex dynamics and behaviors[79]. In future studies, we plan to develop sophisticated algorithms to infer continuous inter-cluster relations within 3D tissue by carefully exploiting spot relations; (2) the prediction accuracy for sample classification in supervised tasks is moderate. This could potentially be enhanced by integrating other modalities in SRT data, including cellular features from histological images[11]; and (3) the core predictions are robust and reproducible across random seeds, despite the inherent variability associated with the stochastic nature of deep learning models. To further improve stability, we will explore ensemble learning and consensus clustering (Supplementary Fig. 33a, b and Supplementary Note 3).

## Methods
### stClinic model

The stClinic model comprises five components (Fig. 1a–d): (1) extracting batch-corrected features from multi-slices with a dynamical graph; (2) evaluating niche importance in clinical outcomes through attention-based supervised learning; (3) transferring labels from reference via zero-shot learning; (4) integrating multi-omics data from the same slice; and (5) aligning data across different slices or technologies, both of which rely on external tools for latent feature initialization.

### Learning shared latent features across multiple slices by dynamic graph learning

stClinic learns batch-corrected features ($\mathbf{z} \in R^{d \times n}$) by aggregating information from dynamically evolving neighboring nodes with similar characteristics within and across slices, where $d$ and $n$ are the dimension size and the number of spots, respectively. Specifically,

**Construction of the initial unified graph across multi-slice.** We initially constructed a unified graph ($\mathbf{G}_0 = (\mathbf{V}, \mathbf{E})$) (Supplementary Fig. 1) to establish links between two spots from multi-slices using omics profiles ($\mathbf{X} = (\mathbf{x}_1, \ldots, \mathbf{x}_L), \mathbf{x}_i \in R^{m \times n_i}$) and spatial location data ($\mathbf{S} = (\mathbf{s}_1, \ldots, \mathbf{s}_L), \mathbf{s}_i \in R^{n_i \times 2}$), where $L$, $m$, and $n_i$ represent the number of slices, common features of all slices, and spots in the $i$th slice, respectively. Intra-edges within a slice were established by measuring the Euclidean distance between spots, retaining an average of 5–6 nearest neighbors based on a predefined threshold $r$ or using the $k$-nearest neighbor method. Inter-edges linking spot pairs from different slices were identified as mutual nearest neighbors (MNN) based on feature similarity using the MNN method[80].

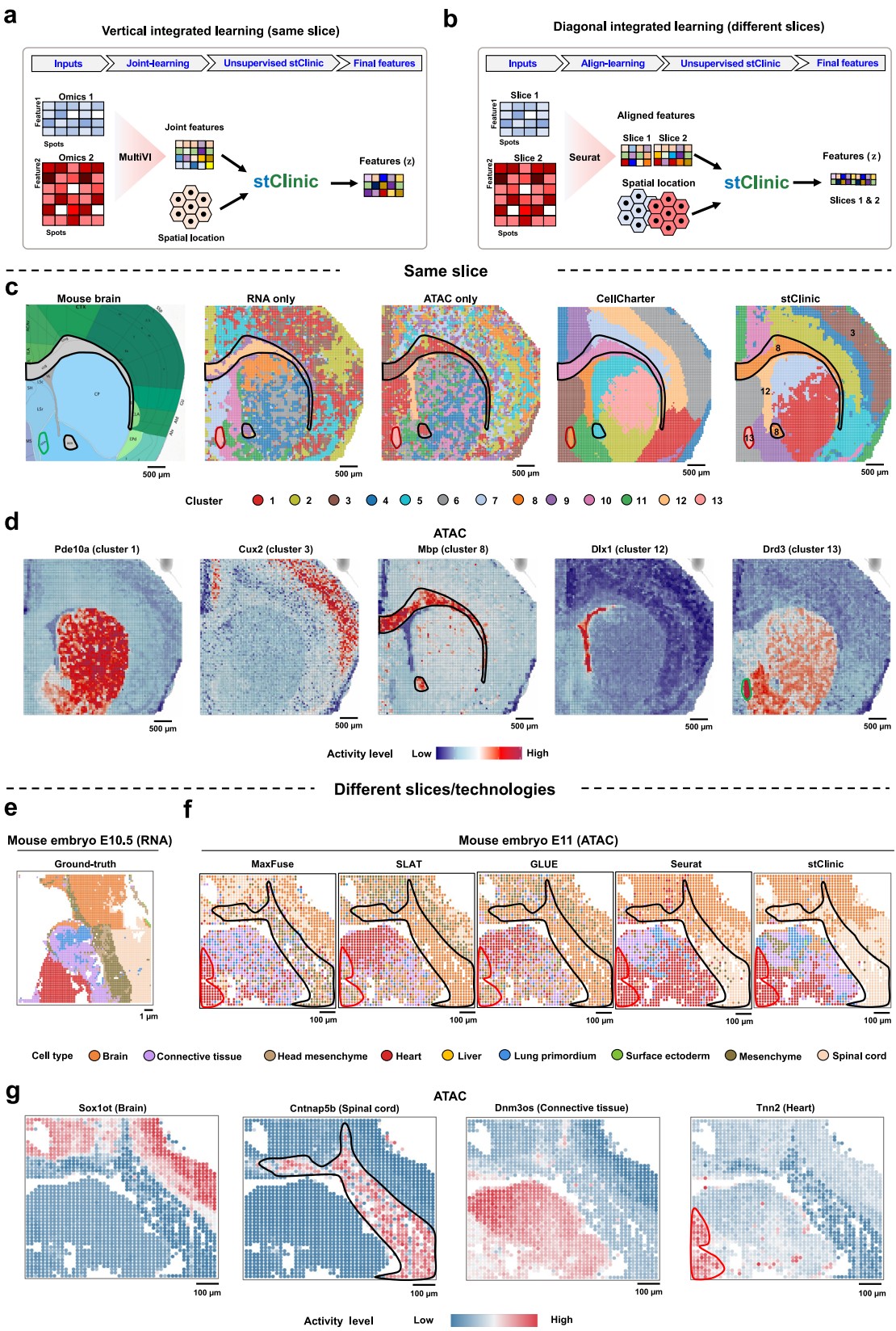

**Encoding features by a dynamically evolving graph model.** We continuously updated the graph model in the following ways: (i) utilizing a VGAE to learn shared features by integrating omics profile data ($X \in R^{m \times n}$) and an adjacency matrix ($A_{it} \in R^{n \times n}$) representing the unified graph $G_{it}$, then employing $L$ slice-specific BatchNorm decoders to reconstruct each omics profile data and adjacency matrix, while incorporating GMM to classify spots into distinct GMM components; and (ii) removing edges between spots from different components in $G_{it}$ to generate $G_{it+1}$, and updating the graph $G_{it}$ by $G_{it+1}$. Note that the initial graph input is $G_0$.

**Fig. 7 | stClinic improves the detection of finer structure by integrating spatial multi-omics data from the same and different slices. a** stClinic learns joint features by integrating latent features from multi-omics tools like MultiVi alongside spatial location data within dynamic graphs. **b** Leveraging aligned features from multi-omics tools like Seurat in a multi-slice integrative condition, stClinic employs the same strategy as to learn the final features. **c** Manual annotation of mouse brain coronal section, and spatial domains identified by single modality (RNA or ATAC), CellCharter, and stClinic. **d** Spatial ATAC levels of marker genes for cluster 1 (*Pde10a*), cluster 3 (*Cux2*), cluster 8 (*Mbp*), cluster 12 (*Dlx1*), and cluster 13 (*Drd3*), for mouse brain coronal section data denoised by stClinic. **e** Cell-type distribution of brain, connective tissue, head mesenchyme, heart, liver, lung primordium, mesenchyme, spinal cord, and surface ectoderm on mouse embryo E10.5 tissue profiled by Stereo-seq. **f** Transferred cell-type distribution on mouse embryo E11 tissue profiled by spatial ATAC-seq using MaxFuse, GLUE, SLAT, Seurat, and stClinic, respectively. **g** Spatial ATAC levels of marker genes for brain (*Sox1ot*), spinal cord (*Cntnap5b*), connective tissue (*Dnm3os*), and heart (*Tnn2*), for mouse embryo E11 data denoised by stClinic. Source data are provided as a Source Data file.

**The VGAE probabilistic model combined with GMM.** Given $K$ clusters, the shared features $\mathbf{z}$ could be obtained through the VGAE via the reparameterization, and $c$ is a categorical variable whose probability is discrete. $p(\mathbf{z}|c)$ is a mixture of Gaussian distributions parameterized by the mean value vector $\mathbf{u}_c$ and the covariance matrix $\boldsymbol{\sigma}_c$ conditioned on $c$. Considering that $\mathbf{X}$, $\mathbf{A}$ and $c$ are independently conditioned on $\mathbf{z}$, then the joint probability $p(\mathbf{X}, \mathbf{A}, \mathbf{z}, c)$ can be factorized as:

$$p(\mathbf{X}, \mathbf{A}, \mathbf{z}, c) = p(\mathbf{X}|\mathbf{z})p(\mathbf{A}|\mathbf{z})p(\mathbf{z}|c)p(c) \tag{1}$$

Each factorized variable defined as follows:

$$c \sim \text{Cat}\left(\frac{1}{K}\right) \tag{2}$$

$$\mathbf{z} \sim \text{N}(\mathbf{u}_c, \boldsymbol{\sigma}_c{}^2 \mathbf{I}) \tag{3}$$

Maximizing the log-likelihood of the observed omics profiling data and the unified graph is intractable, therefore, the evidence lower bound is optimized instead:

$$\begin{aligned} \log p(\mathbf{X}, \mathbf{A}|\mathbf{z}, c) &\geq E_{q(\mathbf{z}, c|\mathbf{X}, \mathbf{A})}\left[\log \frac{p(\mathbf{X}, \mathbf{A}, \mathbf{z}, c)}{q(\mathbf{z}, c|\mathbf{X}, \mathbf{A})}\right] \\ &= \log(p_{\theta 1}(\mathbf{X}|\mathbf{z})) + \delta \log(p_{\theta 2}(\mathbf{A}|\mathbf{z})) \\ &\quad - \varphi \text{D}_{\text{KL}}(q(\mathbf{z}, c|\mathbf{X}, \mathbf{A})||p(\mathbf{z}, c)) \end{aligned} \tag{4}$$

where $\log(p_{\theta 1}(\mathbf{X}|\mathbf{z}))$ encourages the reconstructed data $\mathbf{X}'$ to resemble the input omics profiling data $\mathbf{X}$, and the network $p_{\theta 1}$ indicates $L$ slice-specific BatchNorm decoders. $\log(p_{\theta 2}(\mathbf{X}|\mathbf{z}))$ encourages the reconstructed graph to match the unified graph, which is achieved by an inner product between the features: $\mathbf{A}' = \text{Sigmoid}(\mathbf{z}^T, \mathbf{z})$. KL divergence from the MOG prior $p(\mathbf{z}, c)$ to the variational posterior $q(\mathbf{z}, c|\mathbf{X}, \mathbf{A})$, regularizing the latent features $\mathbf{z}$ to lie on a MOG manifold. $\delta$ and $\varphi$ are used to control the weight of each term. The minimization of $\mathbf{X}$ and $\mathbf{X}'$ can be calculated by the mean square error:

$$L_{Exp} = \frac{1}{L} \sum_{i=1}^{L} ||\mathbf{X}_i - \mathbf{X}_i'||_2 \tag{5}$$

and the minimization of $\mathbf{A}$ and $\mathbf{A}'$ can be calculated by the cross-entropy loss:

$$\mathscr{L}_{Adj} = -\frac{1}{n \times n} \sum_{u=1}^{n} \sum_{v=1}^{n} (a_{uv} \times \log(a_{uv}') + (1 - a_{uv}) \times \log(1 - a_{uv}')) \tag{6}$$

where $a_{uv}$ and $a_{uv}'$ are elements in the $u_{th}$ row and the $v_{th}$ column of the adjacency matrix $\mathbf{A}$ and $\mathbf{A}'$, respectively. Hence, the goal of the probabilistic model is summarized as follows:

$$L_{unsup} = -\mathscr{L}_{ELBO}(\mathbf{X}, \mathbf{A}) = L_{Exp} + \delta \mathscr{L}_{Adj} + \varphi \text{D}_{\text{KL}}(q(\mathbf{z}, c|\mathbf{X}, \mathbf{A})||p(\mathbf{z}, c)) \tag{7}$$

**the overall structure of VGAE.** The specific encoder structure of VGAE can be built by stacking multiple multi-head graph attention (GAT) layer. Specifically, each layer is defined as follows:

$$\mathbf{h}_i^{l+1} = \text{ELU}\left(\frac{1}{Q} \sum_{q=1}^{Q} \sum_{j \in \mathbf{N}_i} a_{ij}^q \mathbf{W}^q \mathbf{h}_j^l\right) \tag{8}$$

$$a_{ij}^q = \frac{\exp(\text{LeakyReLU}((\mathbf{a}^q)^T[\mathbf{W}^q \mathbf{h}_i^l || \mathbf{W}^q \mathbf{h}_j^l]))}{\sum_{o \in \mathbf{N}_i} \exp(\text{LeakyReLU}((\mathbf{a}^q)^T[\mathbf{W}^q \mathbf{h}_i^l || \mathbf{W}^q \mathbf{h}_o^l]))} \tag{9}$$

where $Q$ represents the number of attention heads and the default value of 3, $\mathbf{N}_i$ is the neighbor nodes of the spot $i$, $\mathbf{h}_j^l$ indicates the input features of the node $j$ in the $l$th GAT layer, $\mathbf{W}^q$ is the linear transformation weight matrix for input features in the $qth$ attention head, $a_{ij}^q$ is the normalized attention coefficients calculated by the $qth$ attention head via SoftMax activation. The encoder is composed of two layers of GAT; the dimensions of the first and second layers are 512 and 10, respectively.

Moreover, the one-layer linear decoder specific to the $i$th slice, along with BatchNorm, is used to reconstruct $i$th omics profiling data ($\mathbf{X}_i'$) from the latent feature $\mathbf{z}_i$[36]:

$$\mathbf{X}_i' = \gamma_i \times \frac{\mathbf{h}_i - \mu_i}{\sqrt{\sigma_i^2 + \epsilon}} + \beta_i \tag{10}$$

$$\mathbf{h}_i = \mathbf{W}\mathbf{z}_i + \mathbf{b} \tag{11}$$

where the dimension of $\mathbf{h}_i$ of the same with $\mathbf{X}_i$, $\mu_i$ and $\sigma_i^2$ are the mean and variance of spots in the $i$th slice, $\gamma_i$ and $\beta_i$ are responsible for the slice-specific scaling and shifting parameters, and $\epsilon$ is a constant.

**graph dynamic evolution strategy.** We adopted the following graph evolution strategy to ensure each spot is connected to spots with the most similar characteristics:

i)  Parameter initiation: We pre-trained the VGAE model without GMM regularization for 300 iterations to obtain preliminary features $\mathbf{z}$, with $\mathbf{X}$ and $\mathbf{A}_0$ as input. Subsequently, we used the GMM model to predict the clusters from features $\mathbf{z}$, and utilized the mean ($\mu_c$) and variance ($\sigma_c$) of each cluster to initialize the parameters of GMM distribution; and

ii) Dynamic training: We trained the VGAE model with GMM regularization, using $\mathbf{X}$ and $\mathbf{A}_{it}$ as input to yield features $\mathbf{z}$, where $it$ ranges from 0 to 2; predicted clusters from features $\mathbf{z}$ using the GMM model; removed links between two spots from different clusters in $\mathbf{A}_{it}$ to generate the new adjacency matrix $\mathbf{A}_{it+1}$; and then updated $\mathbf{A}_{it}$ with $\mathbf{A}_{it+1}$.

We iteratively trained the graph model until convergence, and then applied $\mathbf{z}$ for spatial clustering, visualization, and data denoising (Fig. 1b, d).

## Predicting clinically relevant niches by attention-based supervised learning

We proposed a slice representation method that reflects the underlying data structure for each niche per slice across the population, enabling the construction of the associations between niche and clinical information ($\mathbf{Y} = (y_1, \ldots, y_L)^T \in R^{L \times 1}$). Specifically,

(i) after extracting shared features ($\mathbf{z}$) from multi-slices using a dynamic graph model and predicting clusters, UMAP was collectively applied to the $\mathbf{z}$ space across all slices, creating a unified embedding space for comparability between slices and clusters. The $k$th cluster in the $i$th slice was then characterized by six statistical measures: mean ($\boldsymbol{\mu}_{1,i}{}^k$ and $\boldsymbol{\mu}_{2,i}{}^k$), variance ($\boldsymbol{\sigma}^2_{1,i}{}^k$ and $\boldsymbol{\sigma}^2_{2,i}{}^k$), max ($\max_{1,i}{}^k$ and $\max_{2,i}{}^k$), min ($\min_{1,i}{}^k$ and $\min_{2,i}{}^k$) within the two UMAP embeddings, along with the proportion within the $i$th slice ($\mathbf{P}_i{}^k$) and across all other slices ($\mathbf{P}_{o,i}{}^k$);

(ii) inspired by attention-based models emphasizing capturing more critical information to the current task from rich information[11], the $i$th slice representation ($\mathbf{r}_i$) was defined by a vector of clusters, using the 10-dimensional statistical measures via attention by the following formula:

$$\mathbf{r}_i = \sum_{j=1}^{10} \lambda_i^j \cdot \mathbf{f}_i^j \quad (12)$$

$$\lambda_i^j = \frac{\exp((\mathbf{a}_j)^T(\|_{j=1}^{10}\mathbf{f}_i^j))}{\sum_{o=1}^{10} \exp((\mathbf{a}_o)^T(\|_{j=1}^{10}\mathbf{f}_i^j))} \quad (13)$$

where $\mathbf{a}_j \in R^{10K \times 1}$ is the parameter vector of the $j$th statistical measure, and $\mathbf{f}_i^j \in R^{1 \times 10}$ indicate the $j$th statistical measure of the $i$th slice. A higher inner product between $\mathbf{a}_j$ and $\|_{k=1}^{K}\mathbf{f}_i^k$ represents that the role of the $j$th statistical measure is more important to the $i$th slice; and

(iii) an FC layer with SoftMax or Cox layer ($\mathbf{Y}' = F(\mathbf{Wr} + \mathbf{b})$) was employed to predict sample labels, guided by clinical information. If the clinic information is survival time, F indicates the Cox layer[81], the loss function of which is summarized as follows:

$$\mathscr{L}_{COX} = -\sum_{C(i)=1} (\mathbf{y}_i' - \log \sum_{\mathbf{y}_j' \geq \mathbf{y}_i'} \exp(\mathbf{y}_j')) \quad (14)$$

where $\mathbf{y}'$ is the predicted Hazard ratio (HR) value, $C(i) = 1$ represents the non-censored slice set, and only the non-censored slices are taken into the computation of Cox loss. If the clinical information is a category variable (e.g., primary and metastasis), F indicates the SoftMax layer, the loss function of which is summarized as follows:

$$\mathscr{L}_{CLS} = \frac{1}{L}\sum_{l=1}^{L}(-\sum_{i=1}^{p} \mathbf{y}_i \log(\mathbf{y}_i')) \quad (15)$$

where $L$ and $P$ are the numbers of slices and classes, respectively, and $\mathbf{y}_i$ and $\mathbf{y}_i'$ are the label vector of the $i$th slice from the ground truth and prediction.

After the model training, $\mathbf{W}^T$ reflects the significance of each TME in association with clinical information (Fig. 1c, d).

## Datasets and preprocessing

**Spatial omics data.** In our study, we analyzed publicly available spatial omics data from diverse tissues, including human DLPFC, breast cancer, colorectal, and LM samples, as well as mouse brain and embryo samples (Supplementary Table 1). Specifically, (i) the human DLPFC dataset contains 12 slices, with the number of spots ranging from 3460–4789, and a median of 3844[39]; (ii) the 3D Hippo sample includes seven adjacent slices, totaling 10,908 spots[44]; (iii) VIDC, BAS1, and BAS2 slices contain 4727, 3798, and 3987 spots, respectively; (iv) the TNBC sample consists of 43 slices with detailed clinical information, including age, stage, survival time, with the number of spots ranging from 554–3116, with a median of 1264[46]; (v) the CRC and LM dataset comprises 24 tissue sections, including 10 metastatic cancers and 14 primary cancers, with spot counts ranging from 1048 to 4796, and a median of 2636om 1048 to 4796, and a median of 26; (vi) the mouse brain coronal section, profiled using spatial ATAC-RNA-seq technology, contains 9215 spots[67]; (vii) two mouse embryo slices, E10.5 (Stereo-seq) and E11 (Spatial ATAC-seq), have 4132 and 2099 spots, respectively[68,69]; (viii) the E8.75 seqFISH[72] and E9.5 Stereo-seq[69] mouse embryo slices include over 10,000 and 5000 cells/spots, respectively; and (ix) the Visium and Xenium slices from the same breast cancer tissue have 3841 and 100,642 spots/cells, respectively[74].

**scRNA-seq data.** To comprehensively understand the complex structure of inter- and intra-tumoral TMEs across various cancer samples, we estimated the proportions of different cell types using scRNA-seq data with GraphST. For the VIDC and BAS1 slices, we utilized scRNA-seq data from 3961 cells of the CID4535 sample[12] to quantify spatial distributions of 11 cell types: CAFs, perivascular-like (PVL), macrophage, monocyte, endothelial, CD8+ T, CD4+ T, DC, endothelial, cancer epithelial, and plasmablasts. In the TNBC dataset with 43 slices, we used the scRNA-seq data from 42,512 cells across 10 TNBC samples[12], covering 23 cell types. For the CRCLM dataset, we analyzed a scRNA-seq dataset comprising 6275 cells[82] from one CRC patient to infer cell-type proportions for each spot across 29 cell types (Supplementary Table 1).

**SRT data preprocessing.** For each slice of SRT data, we followed the standard workflow of the scanpy package[83], which includes normalization and log-transformation of the raw gene expression. Subsequently, we selected the top 5000 highly variable genes (HVGs) for each slice. The intersection of these HVGs across all slices was considered as common genes, and the horizontal concatenation of these common genes across all spots from multi-slices constituted the input data $\mathbf{X}$. In scenarios involving integrative analysis across multiple heterogeneous slices, we suggested users consider selecting a larger number of HVGs to ensure that the common gene set comprises ~1000 genes.

**Spatial multi-omics data preprocessing.** For spatial multi-omics data from the same slice, we leveraged multiVI to project these profiles into a joint-learning feature space, taking the low-dimensional features as input data $\mathbf{X}$. For spatial multi-omics data from different slices, we transformed these data into a common feature space using Seurat, treating the horizontally concatenated feature matrices of all spots as input data $\mathbf{X}$.

## Selecting the number of clusters in GMM

We adopted the following strategies to choose the number of clusters ($K$) in the GMM model. Specifically, (1) for datasets with detailed histological annotations, such as human DLPFC, mouse brain, and mouse embryo tissues, $K$ is set to the number of different annotation types; and (2) for datasets with generalized annotations, like human BRCA and TNBC tissues, $K$ is determined by the number of the significantly different eigenvalues of $\mathbf{X}^T\mathbf{X}$[84], where $\mathbf{X}$ represents the input data.

## Clustering and visualization

After learning the shared latent features $\mathbf{z}$ from multiple SRT datasets by unsupervised stClinic, we utilized clustering algorithms to predict

clusters based on these features. For the DLPFC dataset with known histological annotations, we employed the GMM model from the mclust package[40] to predict clusters. For heterogeneous tissues like tumor, brain, and embryo, we applied the Louvain algorithm from the scanpy package. The "tl.umap" function from the scanpy package was used to map the features $z$ into a two-dimensional UMAP spaces, and the "pl.umap" function was used to visualize spot embeddings in different domains. Additionally, the "pl.spatial" function was used to visualize clustering results and gene expression patterns for each slice at the spatial level.

### Identification and functional annotation of upregulated genes
We employed the "tl.rank_genes_groups" function from the scanpy package to perform Wilcoxon tests on gene expression data across various spatial domains, identifying upregulated genes in each domain. These genes were then functionally annotated using the DAVID tool (https://david.ncifcrf.gov/tools.jsp).

### Evaluation of clustering
In addition to the ARI[85] and NMI[86] scores to evaluate the clustering performance by comparing the predicted clusters with the ground truth, we adopted additional metrics, ASW, to evaluate the clustering by calculating the similarities of features between spots within the predicted clusters. The silhouette width is used to measure how similar a spot is to its predicted clusters compared to other clusters, and a higher value means that the spot is well-assigned to its cluster, which is defined as follows:

$$SW(i) = \frac{b(i) - a(i)}{\max\{a(i), b(i)\}} \tag{16}$$

where $a(i)$ and $b(i)$ are the average Euclidean distance of the latent features between spot $i$ and other spots in the same cluster, and $i$ to all spots in the near cluster to which $i$ does not belongs, respectively. The average of the silhouette width of all spots is calculated to evaluate the clustering performance.

### Evaluation of batch correction
To comprehensively evaluate the performance of batch correction algorithms, we adopted two different methods: LISI and F1 score[43]. Specifically, (i) LISI, utilizing a fixed perplexity, chooses the nearest neighbors based on local distribution and computes the inverse Simpson's index to measure the diversity, representing the effective number of types in the neighborhood. In iLISI, scores are calculated for batch labels, with a score close to the expected number of batches indicating effective mixing. For cell-type LISI (cLISI), a score nearing 1 indicates pure clusters, while for histological type LISI (hLISI), a score approaching 1 reflects pure histological type annotations; and (ii) F1 score was employed as an overall measure to evaluate cluster purity and slice mixing based on ASW. $ASW_{slice}$ calculates the ASW value using slice labels as groups, while $ASW_{cluster}$ determines the ASW value using cluster labels as groups. The specific formula is defined as follows:

$$F1 = \frac{2(1 - ASW_{slice}')ASW_{cluster}'}{ASW_{cluster}' + (1 - ASW_{slice}')} \tag{17}$$

where $ASW_{slice}' = \frac{1 + ASW_{slice}}{2}$, and $ASW_{cluster}' = \frac{1 + ASW_{cluster}}{2}$. A higher F1 score indicates better slice integration.

### Assessment of clinical or phenotypical prediction
In the prognostic prediction task, we adopted the median of the cross-validated C-Index[81] to evaluate the agreement between the patient ranking based on the predicted HR and the ranking by survival time. A higher C-Index reflects more accurate survival

predictions, with a value of 0.5 indicating random prediction and 1 indicating complete consistency with the actual observations. For categorical clinical information such as primary and metastasis, we employed the cross-validated classification accuracy and AUCROC value to evaluate the classification performance of stClinic.

### Statistics and reproducibility
No statistical methods were employed to predefine the sample size. Neither biological nor technical replicates were performed on the biological samples outlined in Figs. 4d and 5d. All data were sourced from the public domain, and no exclusions were made from the analysis. The experiments were not randomly conducted, and the researchers were blinded to allocation during the experiment and assessment of results. Further details can be found in the Reporting summary file.

### Reporting summary
Further information on research design is available in the Nature Portfolio Reporting Summary linked to this article.

## Data availability
All datasets used in this manuscript are publicly available. The DLPFC dataset is available from the R package spatialLIBD (http://spatial.libd.org/spatialLIBD/)[39]. The 3D hippo dataset can be accessed via the link (https://drive.google.com/drive/folders/10lhz5VY7YfvHrtV40Mwaq LmWz56U9eBP?usp=sharing). The VIDC, BAS1, and BAS2 datasets of human Luminal B breast cancer are available on the 10X Genomics Website (https://www.10xgenomics.com/datasets/). The Visium and Xenium datasets of human HER2+ breast cancer are available from the Gene Expression Omnibus (GEO) with GSE243280. The seqFISH and Stereo-seq datasets of mouse embryo are available at (https://marionilab.cruk.cam.ac.uk/SpatialMouseAtlas/) and (https://db.cngb.org/stomics/mosta/download/), respectively. The TNBC dataset, containing 43 slices, is available from GEO with GSE210616, and the corresponding clinical information is available at the website (https://doi.org/10.1158/0008-5472.CAN-22-2682). The CRCLM dataset can be accessed via the link (https://drive.google.com/file/d/1QsQIT0-iwcWBFzUBcLUPKYuSnSBxfaME/view?usp=drive_link). The Brain dataset profiled by spatial ATAC-RNA-seq is accessible from GEO with GSE205055. The E10.5 mouse embryo profiled by Stereo-seq and the E11 mouse embryo profiled by Spatial ATAC-seq are available from the link (https://db.cngb.org/stomics/mosta/download/) and GEO with GSE171943, respectively. The scRNA-seq datasets of CID4535, TNBC, and CRC samples are available from GEO under accession numbers GSE176078, GSE176078, and GSE132465, respectively. Source data provided for this paper are available at figshare (https://doi.org/10.6084/m9.figshare.27376827)[87].

## Code availability
stClinic and all the code for reproducing the analyses and benchmarking are freely available under the MIT License. stClinic is implemented based on Python 3.8.5 and R 4.3.2. Other tools and packages used in the data analysis include: anndata 0.9.2, numpy 1.22.3, pandas 2.0.3, scipy 1.10.1, matplotlib 3.7.2, scanpy 1.9.3, umap-learn 0.5.3, louvain 0.8.1, h5py 3.9.0, torch 2.4.0, torchaudio 2.4.0, torchvision 0.19.0, tqdm 4.65.0, hnswlib 0.5.1, rpy2 3.5.1, scikit-learn 1.3.0, scikit-misc 0.2.0, seaborn 0.11.2, lifelines 0.27.8, network 3.1, torch-sparse 0.6.18, torch-scatter 2.1.2, Squidpy 1.2.2, SEDR 1.0.0, STAligner 1.0.0, GraphST 1.1.1, Stitch3D 1.0.3, PRECAST 1.6.4, SLAT 0.3.0, BANKSY 0.99.12, SpaGCN 1.2.7, MultiVI 1.3.0, scVI 1.3.0, peakVI 1.3.0, CellCharter 0.3.4, GLUE 0.3.2, MaxFuse 0.0.2, stClinic 0.0.10, Seurat v4, ggplot2 3.3.6. The codes are publicly available at Zenodo https://zenodo.org/records/15246396[88]. The stClinic tool will be maintained and updated at https://github.com/cmzuo11/stClinic.

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

## Acknowledgements

This work was supported by the National Natural Science Foundation of China (Nos. T2341007, 12131020, T2350003, 42450135, 42450084, 12326614 and 12426310 to L.N.C., Nos. 32300523 and 62132015 to C.M.Z., Nos. T2350010 and W2431059 to Y.X., and No. 62072212 to Y.W.), the Zhejiang Province Vanguard Goose-Leading Initiative (No. 2025C01114 to L.N.C.), Fundamental Research Funds for the Central Universities, JLU to C.M.Z., Science and Technology Commission of Shanghai Municipality (No. 23JS1401300 to L.N.C.), JST Moonshot R&D (No. JPMJMS2021 to L.N.C.), the Open Project of Shanghai Collaborative Innovation Center of Endoscopy Fudan University to C.M.Z., and the Development Project of Jilin Province of China (No. 20220508125RC to Y.W.). Selected images were adapted from Servier Medical Art, under the Creative Commons Attribution 4.0 International License.

## Author contributions

C.M.Z. conceived and designed the study. C.M.Z. and J.J.X. implemented the model and performed the experiments, with the assistance of Y.P.X. C.M.Z. wrote the manuscript with feedback from all authors. Y.X., P.T.G., J.Z. and Y.W. analyzed the experiment results. C.M.Z. and L.N.C. co-supervised the study. The authors read and approved the final manuscript.

## Competing interests

The authors declare no competing interests.
