## [Peer Review File · Nature Communications]

stClinic dissects clinically relevant niches by integrating spatial multi-slice multi-omics data in dynamic graphs

Corresponding Author: Dr Chunman Zuo

Editorial Note: Figures on page 7, 35, 39, 48 and 57 in this Peer Review File has been amended to remove third-party material where no permission to publish could be obtained.

Version 1:

Reviewer comments:

Reviewer #1

(Remarks to the Author)

This paper introduces a tool for integrating multi-slice spatial omics data. When integrating same omics data, an iteratively pipeline involving VGAE and dynamically evolving graphs is used. When integrating multi-omics data, the pipeline takes input from processed latent features from other multi-omics integration tools like MultiVI. While the computational approach in the integration part seems standard, the computational utilities for the clinical predictions are relatively new and differ from other existing methods. I mainly have some concerns on the rationale of clinical prediction part and comparisons to other related methods.

Specific points:

1. On line 8 of page 19, it was mentioned that the k th cluster in the i th slice was quantified by a vector of six statistical measures on two UMAP embeddings of its latent features. Was UMAP applied to the multislice data at once or individually on each slice or cluster? If it is the latter case, why are the UMAP embeddings comparable? Also, UMAP is translational and rotational invariant. The features like mean, variance, and max/min on each UMAP component is not invariant under rotation. This point needs to be justified.
2. In the comparison to other methods presented in Fig. 2, the mclust algorithm was used for every method. Was this the clustering algorithm used in the original works of other methods? Different methods may have different corresponding suitable clustering algorithms. It would be more appropriate use the results in the original publications of these methods, as the DLPFC dataset was used in all those studies.
3. How was the number of clusters chosen for the Gaussian Mixture Model?
4. All applications presented to spatial transcriptomics data were performed on Visium data. Could you also test the method on ST datasets from different technologies, like the examples presented in the SLAT paper?
5. Related to point one, it is surprising to me that the statistics on individual UMAP embedding components work so well on predicting clinical outcomes. Could you further clarify the biological and clinical interpretation of these statistics? The stability of the findings should also be evaluated. For example, when performing UMAP and the VGAE with different random seeds, do you still observe roughly the same group of spots identified to be important as the results in Fig. 4?
6. The code seems well written with easy-to-understand documentation. I would suggest to make the package installable from PyPI for easier installation and better usability.
7. It should be stated more clearly in abstract, introduction, and the method overview, that when integrating multi-omics data, stClinic relies on other packages for initializing the common features across different omics. I find the current presentation a little misleading to state that stClinic can use latent features from other tools rather than stClinic has to use latent features from other tools.

(Remarks on code availability)

Tutorials for reproducing the main results are provided. As a minor comment, I suggest to make the package available on PyPI for ease of use.

Reviewer #2

(Remarks to the Author)

This study introduced stClinic, a computational framework for integrating spatial multi-slice multi-omics data. stClinic includes multiple modules, such as homogenous and heterogenous alignment, detection of inter- and intra-tumoral niches, identification of clinically relevant niches, and label transferring, all of which were demonstrated through examples and

benchmarking with existing tools. With the increasing volume and modality of the spatial data in the field, there is an unmet need for robust methods to effectively integrate spatial multi-omics data from the same patients across consecutive tissue sections, as well as across patients and cohorts. In this context, stClinic is potentially interesting, and the integration results produced by stClinic look promising. However, multiple major weaknesses are also noted (see major comments below) and substantial revisions and improvements are needed.

Major comments:

1. The algorithm requires an adjacency matrix as an input, but oftentimes this information would not be available. How will the algorithm cope with this situation? Please provide some examples of applications in such cases and discussions.
2. The author performed a comparative analysis on adjacent slices of three samples (Fig. 2) and stated that "stClinic exhibited higher consistence in annotation than all other methods" To evaluate the performance of the annotation, it is not sufficient to simply quantify the consistency of annotations across the adjacent slices for each platform. The best annotation should be able to capture the spatial heterogeneity (changes in tissue structures across adjacent slices) precisely and accurately, as tissue structures may vary across consecutive slices. It is therefore important to assess the accuracy of the annotations against the ground truth for each slice. Such assessment was not performed in this study. Additionally, the ground truth here is unclear. The best ground truth would be high-resolution, spot-level, or pixel level pathology annotations of the paired H&E images.
3. For the evaluation of batch-effect correction (Fig. 2), ST data generated from different slides and batches are preferred. However, it is unclear whether the ST data derived from these adjacent slices in Fig. 2 were from the same slide or batch. If that is the case, the authors should include additional ST data from a larger number of adjacent slices (>10 or at least >5 adjacent slices from a sample) to better evaluate the performance of stClinic. Such public datasets currently exist and could provide a more comprehensive assessment.
4. Importantly, to determine if a cluster from one slice of a sample is identical or highly similar to another slice of the same sample or a different sample, the statistical assessment shown in Fig. 2E is insufficient. The morphological and histological similarity, as well as transcriptomic similarity, should also be evaluated to ensure that the batch correction was not under-corrected or overcorrected.
5. From Supplementary Figure 1a, stClinic did not outperform PRECAST and some other methods when considering ASW, F1, cLISI, and iLISI. Additionally, from Supplementary Figures 1b, 2a, 3a, and Figure 2d, the clustering separations were much clearer in PRECAST than in stClinic. A bar plot to show the slice compositions in each cluster from each method could be a better way.
6. To illustrate that stClinic can discern heterogeneous niches from complex disease tissues, this study analyzed ST data generated on breast cancer and hepatocellular carcinoma (Fig. 3). There are obvious limitations with the ST datasets utilized for analysis. The size of the tissues is overall small. The H&E images are low-resolution and somewhat blurry; most importantly, the pathology annotations are not at the spot level, the annotations are oversimplified and lack of essential granularity. This is clearly an issue. For example, large areas in panel a are labelled as "invasive cancer" while there are clearly non-neoplastic tissues within those regions; while in panel g, the complex tissues are labeled as "stroma" and "tumor/normal epithelium, which is overly simplified and not acceptable, as tumor/normal epithelium are mixed up in annotations and lack of essential granularity. It is therefore hard to trust the results in Fig. 3 and determine the spatial domains from which tool align better with the pathology annotations. This is a major weakness of the study. The authors should select more appropriate ST datasets to ensure a robust assessment. Additionally, other approaches for spatial domain analysis such as Seurat, SpaGCN, and several recently published tools, should also be considered in the benchmarking.
7. In Fig. 3d, some clusters, such as clusters 1, 3, 7, showed very inconsistent cell compositions between the two samples (CID4465 and CID 44971). In Fig. 3i, cluster 9 has multiple cell components and its functions are not clear. It is a bit confusing why the author specifically emphasizes this cluster.
8. Figure 4a is confusing. If the risk group is defined based on survival time, the survival difference will surely be significant. If so, this analysis is not really useful. Additionally, it seems that the definitions of high vs. low risk are different in Figs. 4a and 4f. This is a bit confusing as well. The author should state this clearly in the results or use different naming to distinguish between them.
9. In Fig. 4, the integration results look promising. However, it appears that the dense lymphoid infiltrate in panel d was not identified as a unique cluster in panel e and seems to be mixed with fibrosis, which is a very different tissue structure. What is the likely reason for missing this unique structure, leading to the mixing up of these tissue structures with distinct expression profiles?
10. The findings of cluster 6 and cluster 10 are interesting. However, it is too general to summarize cluster 6 and cluster 10 as malignant and non-malignant niches since cluster 10 also contains non-malignant cells. Given that cluster 6 showed high expression of multiple immune signals, it would be beneficial to identify the major cell types and TME structures it captures. Additionally, further analysis/discussions are needed to provide insights into how these findings may translate to patient survival.

11. For lymphocyte aggregates in the BC sample, the gene expression was diffused across the whole slide, especially IGHG1 and IGKC. Lymphocyte aggregates cannot be directly revealed from these genes. Are there any other more distinguishable markers? Please also provide zoom-in H&E images for these regions in parallel in Supplementary Figure 5c.

12. In Figure 5i, CRC liver metastasis and liver cancer are substantially different. Please further validate the signature in CRC cohorts. It would also be helpful to see if the signature can predict metastasis of CRC in independent cohorts.

13. The conclusion from Figure 5 that cluster 1 may represent the initial cell populations infiltrating normal liver tissue does not have enough evidence to be supported, and the definition of metastatic seeding clone will need genomic-level evidence.

14. In Figure 5g, the signals of some markers are diffused, such as FCGR3A and proliferative markers. Additionally, some markers, like SPP1, are not specifically enriched in the cluster 1 region. Given this, it is challenging to assert that SPP1+ myeloid cells may be uniquely enriched and play a role in this cluster. The analysis of the functions of this cluster needs significant improvement.

15. Lastly, for the label transfer function of stClinic, it would be beneficial to show an example from tumor samples. Tumors generally have a higher degree of heterogeneity, making the label transfer task more challenging. However, demonstrating this is very important for the application. Benchmarking with other published software is needed. For multi-omic data integration, it might be helpful to also benchmark with MaxFuse, a recently published software aimed at diagonal integration.

Minor comments:

1. In page 2, lines 22-23, more related studies should be listed to support why certain cellular niche could be clinically relevant. What makes studying cellular niches more important compared to studying individual TME cell states alone?

2. In page 3, lines 12-14, the author listed 3 published computational approaches but only pointed out the limitations of PASTE. What about the other two? Please add a bit more discussion on their limitations. Additionally, provide a definition of homogeneous/heterogeneous slices integration for the readers' understanding. How do they differ from each other in concept, and how do they differ from (iii)?

3. For Fig. 1, the author needs to provide the definitions of vertical and diagonal integration in the corresponding text for the readers' understanding. Fig. 1 is too crowded and hard for the readers to capture the key points. Please simplify it.

4. In Fig. 2d-f, the degree of cluster mixing and the number of clusters depend on the parameter settings of each tool.

5. In Fig. 3, when looking at the results in detail, cluster 4 of stclinic in panel c does not seem to match the locations, numbers, and sizes of lymphocyte aggregates in panel a.

6. In page 7, line 4-16, and page 8, lines 1-18, please cite the figure panels right after each point, instead of putting all the figure panels to the end. The citation of Figure 3 panels is not in order. Also, Figure 3a and g showed pathology annotations in different ways, please make them consistent.

7. Please provide the DEG list of clusters defined by stClinic, and add the key genes shown in Fig. 3e to the dotplot in Supplementary Figure 6.

(Remarks on code availability)

the webpage lacks detailed documentation for uses to run the tool and interpret the results

Reviewer #3

(Remarks to the Author)

(Remarks on code availability)

Version 2:

Reviewer comments:

Reviewer #1

(Remarks to the Author)

All my comments have been addressed in the revised manuscript.

(Remarks on code availability)

Reviewer #2

(Remarks to the Author)

The authors have addressed most of the comments and made improvements to the manuscript. However, some key concerns remain unaddressed:

Original Comment #3: The performance of stClinic should be evaluated in additional samples and cancer types beyond those presented in the previous study (Ref #7). The tissue and annotations from Ref #7 are relatively simple and do not reflect the complexity of real-world tissue contexts. This limitation undermines the generalizability of the tool's performance.

Original Comments #7 and #16: For the newly added breast cancer tissue, the pathology annotations lack essential granularity, making them unsuitable as ground truth for evaluating the performance of different tools. Furthermore, the benchmarking is incomplete, as other highly ranked tools, such as BANKSY (Singhal et al., Nature Genetics 2024), have not been included. The authors removed the tool SPaGCN from the results; however, the rationale provided for its removal is unconvincing. Additionally, the performance of these tools may be influenced by the resolution of clustering analysis (i.e., the number of clusters) and parameter settings.

Marker Issues: The markers LYZ and CD74 are not unique to macrophages, and similarly, IGKC and IGHG3 are not unique markers for memory B cells. This issue raises concerns about the accuracy of cell type annotations.

CNVs Plot in Fig. 4 (Panel K): The copy number variation (CNV) plot appears very noisy. The authors should consider employing alternative methods for CNV inference to improve the clarity and reliability of this analysis.

Original Comment #10: The response to this comment is unsatisfactory and highlights potential limitations of the tool. This issue should be revisited to ensure the tool's robustness.

Original Comment #11: Naming clusters as "tumor-promoting" and "tumor-suppressive" is inappropriate without supporting functional data. Additionally, the gene signatures used in Supplementary Figure 16a need further refinement to ensure their uniqueness and accuracy.

Overall, while the authors have made progress in addressing many of the earlier comments, the issues outlined above require further attention to strengthen the rigor, robustness, and interpretability of the manuscript.

(Remarks on code availability)

Reviewer #3

(Remarks to the Author)

(Remarks on code availability)

Version 3:

Reviewer comments:

Reviewer #2

(Remarks to the Author)

The authors have done an excellent job addressing my remaining comments and have made further improvements to the manuscript. Their commitment to excellence is commendable.

(Remarks on code availability)

Reviewer #1: (Expert in spatial and single-cell transcriptomics, computational and statistical models, and machine learning)

Comment #1: This paper introduces a tool for integrating multi-slice spatial omics data. When integrating same omics data, an iteratively pipeline involving VGAE and dynamically evolving graphs is used. When integrating multi-omics data, the pipeline takes input from processed latent features from other multi-omics integration tools like MultiVI. While the computational approach in the integration part seems standard, the computational utilities for the clinical predictions are relatively new and differ from other existing methods. I mainly have some concerns on the rationale of clinical prediction part and comparisons to other related methods. **Response:** Yes, we thank the reviewer for the comment and encouragement.

Comment #2: On line 8 of page 19, it was mentioned that the k th cluster in the i th slice was quantified by a vector of six statistical measures on two UMAP embeddings of its latent features. Was UMAP applied to the multislice data at once or individually on each slice or cluster? If it is the latter case, why are the UMAP embeddings comparable? Also, UMAP is translational and rotational invariant. The features like mean, variance, and max/min on each UMAP component is not invariant under rotation. This point needs to be justified.

Response: We thank the reviewer for the comment. UMAP was collectively applied to the multi-slice feature space, creating a shared embedding space that ensures comparability across slices and clusters. While UMAP is translationally and rotationally invariant, we calculated statistical measures such as mean, variance, and max/min after embedding to capture niche variations across slices, which remain consistent despite UMAP's invariance. In light of the comment, we have added further clarification regarding the six statistical measures in the UMAP space and revised the Fig. 1c on pages 20, and 34 (in the revised version).

[Figure Redacted]

phenotype prediction, transferring labels from the reference through zero-shot learning, and annotating labels across different types of omics datasets.

Comment #3: In the comparison to other methods presented in Fig. 2, the mclust algorithm was used for every method. Was this the clustering algorithm used in the original works of other methods? Different methods may have different corresponding suitable clustering algorithms. It would be more appropriate use the results in the original publications of these methods, as the DLPFC dataset was used in all those studies.

Response: We apologize for the oversight in describing the clustering methods. In our benchmark comparison, we used the clustering algorithms originally employed by each method. In light of the comment, we have summarized these clustering algorithms in Supplementary Table 2, and revised the corresponding text on pages 6, 8, and 9.

Dataset	Method	Clustering method	Pipeline
DLPFC	SEDR	mclust	https://sedr.readthedocs.io/en/latest/Tutorial3_Batch_integration.html
	GraphST	mclust	https://deepst-tutorials.readthedocs.io/en/latest/Tutorial%20Vertical%20Integration.html
	Stitch3D	Gaussian Mixture Model	https://stitch3d-tutorial.readthedocs.io/en/latest/tutorials/DLPFC/STitch3D_DLPFC.html
	PRECAST	Built-in spatial factor model	https://feiyong.github.io/PRECAST/articles/PRECAST.DLPFC4.html
	STAligner	mclust	https://staligner.readthedocs.io/en/latest/Tutorial_DLPFC.html
	stClinic	mclust	https://github.com/cmzuo11/stClinic/
	SLAT	Leiden	https://slat.readthedocs.io/en/latest/tutorials/cross_technology.html

Tumor samples	Seurat	Louvain	https://satijalab.org/seurat/articles/spatial_vignette
	SEDR	mclust	https://sedr.readthedocs.io/en/latest/Tutorial3_Batch_integration.html
	STAligner	Louvain	https://staligner.readthedocs.io/en/latest/Tutorial_embryo.html
	PRECAST	Built-in spatial factor model	https://feiyong.github.io/PRECAST/articles/PRECAST.BreastCancer.html
	stClinic	Louvain	https://github.com/cmzuo11/stClinic/

Supplementary Table 2. Summary of clustering methods used for comparison across different datasets.

Comment #4: How was the number of clusters chosen for the Gaussian Mixture Model?

Response: We thank the reviewer for the comment. In light of the comment, we have clarified the selection of the number of clusters (K) for the Gaussian Mixture Model. Specifically, (1) for datasets with detailed histological annotations, such as human DLPFC, mouse brain, and mouse embryo tissues, K is set to the number of different annotation types; and (2) for datasets with generalized annotations, like human BRCA and TNBC tissues, K is determined by the number of the significantly different eigenvalues of $X^T X$ ¹, where X represents the input omics data. The updated text can be found on pages 5, and 23.

1 Kiselev, V. Y. *et al.* SC3: consensus clustering of single-cell RNA-seq data. *Nat Methods* **14**, 483-486 (2017).

Comment #5: All applications presented to spatial transcriptomics data were performed on Visium data. Could you also test the method on ST datasets from different technologies, like the examples presented in the SLAT paper?

Response: Yes, we have conducted integrative analysis of ST datasets from different technologies, including i) two mouse embryo slices—seqFISH (E8.75) with over 10,000 cells

and 350 genes ², and Stereo-seq (E9.5) with around 5,000 cells and over 20,000 genes ³; and ii) two breast cancer slices from the same tissue—10X Xenium with 100,642 cells and 313 genes, and 10X Visium with 3,841 spots and over 20,000 genes ⁴.

i) In the embryo datasets, stClinic successfully identified the Otocyst at both E8.75 and E9.5 stages, marked by key genes such as *Gbx2*, *Dll3*, *Lfng*, and *Fst* ⁵. Notably, the Otocyst was not detected by other methods or prior analyses ^{2,3}. These results are shown in Supplementary Fig.28a-e.

ii) In the breast cancer datasets, stClinic demonstrated superior performance in aligning identical niches between two slices. It identified a greater diversity of cancer cell-states, notably detecting four clusters within the ductal carcinoma in situ regions (DCIS #1 and DCIS #2) in the Xenium sample, whereas other methods detected only two or three clusters. These findings were further validated by differential gene expression analysis and cell-type enrichment. The results are shown in Supplementary Fig.29a-e.

In light of the comment, we have incorporated the analysis results and data descriptions on pages 15, 16, and 22.

Supplementary Figure 28. Method comparison for integrative analysis of E8.75 seqFISH and E9.5 Stereo-seq mouse embryo samples. **a** Annotations of seqFISH and Stereo-seq from previous studies ^{2,3}. **b** Spatial domains identified by PRECAST, SLAT, Seurat, and stClinic for both samples. **c** UMAP visualization of features by PRECAST, SLAT, and stClinic, respectively. For each method, the left panel shows color-coded slices, while right panel displays color-coded clusters. **d** Spatial distribution of cluster 12 (Otocyst) in both samples. **e** Violin plots showing gene expression of *Gbx2*, *Dll3*, *Lfng*, and *Fst* across different clusters. Source data are provided as a Source Data file.

Supplementary Figure 29. Method comparison for integrative analysis of Visium and Xenium slices from the same breast cancer tissue. **a** Captured regions of the Visium and Xenium slices on the H&E image. **b** Spatial domains identified by PRECAST, SLAT, Seurat, and stClinic for both slices. **c** UMAP visualization of features by PRECAST, SLAT, Seurat, and stClinic, respectively. For each method, the left panel shows color-coded slices, while right panel displays color-coded clusters. **d** Dot plot showing the expression levels of marker genes across different niches in the DCIS region. **e** Heatmap displaying the relative enrichment of different cell-types across four niches. Source data are provided as a Source Data file.

- 2 Lohoff, T. *et al.* Integration of spatial and single-cell transcriptomic data elucidates mouse organogenesis. *Nature biotechnology* **40**, 74-85 (2022).
- 3 Chen, A. *et al.* Spatiotemporal transcriptomic atlas of mouse organogenesis using DNA nanoball-patterned arrays. *Cell* **185**, 1777-1792. e1721 (2022).
- 4 Janesick, A. *et al.* High resolution mapping of the tumor microenvironment using integrated single-cell, spatial and in situ analysis. *Nat Commun* **14**, 8353 (2023).
- 5 Durruthy-Durruthy, R. *et al.* Reconstruction of the mouse otocyst and early neuroblast lineage at single-cell resolution. *Cell* **157**, 964-978 (2014).

Comment #6: i) Related to point one, it is surprising to me that the statistics on individual UMAP embedding components work so well on predicting clinical outcomes. Could you further clarify the biological and clinical interpretation of these statistics? ii) The stability of the findings should also be evaluated. For example, when performing UMAP and the VGAE with different random seeds, do you still observe roughly the same group of spots identified to be important as the results in Fig. 4?

Response: We thank the reviewer for the insightful comment. We have done the following for each issue. Specifically,

i) Yes, we have clarified that the strong predictive power of the statistics derived from UMAP embeddings comes from UMAP's ability to preserve both local and global structures within batch-corrected features, ensuring consistent cluster alignment across slices. Features

such as cluster presence, proportion, and spatial distribution help characterize key patterns. By summarizing UMAP embeddings with statistics like mean, variance, and extremes, we quantitatively capture these patterns and correlate them with clinical outcomes. While UMAP is translationally and rotationally invariant, the relative positioning in this space retains important biological variability, allowing us to identify clusters that predict clinical outcomes. In light of the comment, we have added related text on page 15.

ii) Yes, we have assessed the stability of the clustering results by running VGAE with different random seeds and comparing the outcomes with those from another graph-based model, STAligner. As anticipated, increasing the number of clusters introduced some variability in cluster assignments across different random seeds, indicating that the model's results can be influenced by seed selection.

To specifically evaluate the consistency of our key findings—cluster 10 (highest positive weight) and cluster 6 (lowest negative weight)—we repeated the VGAE and UMAP processes five times with different random seeds. On average, 75% of the spots in cluster 10 and 92% of the spots in cluster 6 were consistently identified as positive and negative spots, respectively. These results demonstrated that, while some variability is expected due to the stochastic nature of deep learning models, the core predictions of stClinic remain robust and reproducible across different random seeds. In future work, we plan to implement advanced techniques such as ensemble learning and consensus clustering to further enhance the stability and reproducibility of our models, reducing the impact of random seed variability and improving the overall robustness of our findings. The results are shown in Supplementary Fig.31a and b. In light of the comment, we have incorporated the related text on pages 16, and 17.

Supplementary Figure 31. Evaluation of prediction robustness on the TNBC dataset. a Heatmap showing the percentage of interactions between clusters predicted by latent features generated with different seeds (using stClinic and STAligner) and those predicted by latent features (as shown in Fig.4b and Supplementary Fig.8a), across different numbers of clusters. **b** Heatmap displaying the percentage of spots predicted as positive in cluster 10 and negative in cluster 6 across five experiments with different seeds. Source data are provided as a Source Data file.

Comment #7: The code seems well written with easy-to-understand documentation. I would suggest to make the package installable from PyPI for easier installation and better usability.

Response: We thank the reviewer for the suggestion. We have packaged the tool as a Python library, allowing users to easily install it with the command: `pip install stClinic`.

Comment #8: It should be stated more clearly in abstract, introduction, and the method overview, that when integrating multi-omics data, stClinic relies on other packages for initializing the common features across different omics. I find the current presentation a little misleading to state that stClinic can use latent features from other tools rather than stClinic has to use latent features from other tools.

Response: Yes, we have revised the abstract, introduction, and method overview on pages 2, 4, 6, 15, 17, 45, and 46 to clearly state that stClinic relies on latent features from other tools for effective integration across multiple-omics datasets, as suggested.

Comment #9: Tutorials for reproducing the main results are provided. As a minor comment, I suggest to make the package available on PyPI for ease of use.

Response: We appreciate the reviewer's suggestion, and we have made the stClinic package available on PyPI (<https://pypi.org/project/stClinic/>) to improve ease of use.

Reviewer #2: (Expert in cancer genomics, spatial and single-cell transcriptomics, tumour microenvironment and immunology)

Comment #1: This study introduced stClinic, a computational framework for integrating spatial multi-slice multi-omics data. stClinic includes multiple modules, such as homogenous and heterogenous alignment, detection of inter- and intra-tumoral niches, identification of clinically relevant niches, and label transferring, all of which were demonstrated through examples and benchmarking with existing tools. With the increasing volume and modality of the spatial data in the field, there is an unmet need for robust methods to effectively integrate spatial multi-omics data from the same patients across consecutive tissue sections, as well as across patients and cohorts. In this context, stClinic is potentially interesting, and the integration results produced by stClinic look promising. However, multiple major weaknesses are also noted (see major comments below) and substantial revisions and improvements are needed.

Response: Yes, we thank the reviewer for the comment and encouragement.

Comment #2: The algorithm requires an adjacency matrix as an input, but oftentimes this information would not be available. How will the algorithm cope with this situation? Please provide some examples of applications in such cases and discussions.

Response: We thank the reviewer for the comment. In light of the comment, we have clarified the construction approach of the adjacency matrix and added a figure for better illustration. Specifically, for intra-slice edges, stClinic defines neighboring cells/spots based on a specified radius or the k-nearest neighbor method. For inter-slice edges, stClinic identifies neighboring cells/spots as mutual nearest neighbors (MNN) based on feature similarity using the MNN method ⁶. The workflow of this process is provided in Supplementary Fig.1. Additionally, we have incorporated the relevant text as suggested on pages 5 and 17.

Supplementary Figure 1. Construction of the adjacency matrix (unified graph). For intra-slice edges, stClinic defines neighbors based on a specified radius or the k-nearest neighbor method. For inter-slice edges, neighboring cells/spots are identified as mutual nearest neighbors (MNN) based on feature similarity. The adjacency matrix is formed by combining these two types of edges.

6 Haghverdi, L., Lun, A. T. L., Morgan, M. D. & Marioni, J. C. Batch effects in single-cell RNA-sequencing data are corrected by matching mutual nearest neighbors. *Nature Biotechnology* **36**, 421-427, doi:10.1038/nbt.4091 (2018).

Comment #3: The author performed a comparative analysis on adjacent slices of three samples (Fig. 2) and stated that “stClinic exhibited higher consistence in annotation than all other

methods”. To evaluate the performance of the annotation, it is not sufficient to simply quantify the consistency of annotations across the adjacent slices for each platform. The best annotation should be able to capture the spatial heterogeneity (changes in tissue structures across adjacent slices) precisely and accurately, as tissue structures may vary across consecutive slices. It is therefore important to assess the accuracy of the annotations against the ground truth for each slice. Such assessment was not performed in this study. Additionally, the ground truth here is unclear. The best ground truth would be high-resolution, spot-level, or pixel level pathology annotations of the paired H&E images.

Response: We thank the reviewer for the comment and apologize for the lack of clarity regarding the annotation accuracy metrics. In light of the comment, we have clarified that, for each of the 12 slices from three samples, we used six (or four) layers and white matter (WM) annotations from a previous study⁷ as the ground truth to evaluate cluster consistency. To assess annotation accuracy, we employed both the adjusted rand index (ARI) and normalized mutual information (NMI) to measure the similarity between the predicted cluster labels and the ground truth. As shown in Fig.2b and e, and Supplementary Fig.2a, our analysis indicates that stClinic consistently achieves higher agreement with the ground truth compared to other methods. We have revised the related text on pages 6, and 7 to reflect these clarifications.

7 Maynard, K. R. *et al.* Transcriptome-scale spatial gene expression in the human dorsolateral prefrontal cortex. *Nature neuroscience* **24**, 425-436 (2021).

Comment #4: For the evaluation of batch-effect correction (Fig. 2), ST data generated from different slides and batches are preferred. However, it is unclear whether the ST data derived from these adjacent slices in Fig. 2 were from the same slide or batch. If that is the case, the authors should include additional ST data from a larger number of adjacent slices (>10 or at least >5 adjacent slices from a sample) to better evaluate the performance of stClinic. Such public datasets currently exist and could provide a more comprehensive assessment.

Response: We appreciate the reviewer for the insightful comment. In light of the comment, we have clarified that the 12 slices of ST datasets presented in Fig.2 were derived from different batches across three samples, highlighting the efficiency of stClinic in batch-effect correction.

Additionally, we conducted further experiments on an independent ST data consisting of seven consecutive sections profiled by Slide-seq⁸. Our comparative analysis showed that stClinic more accurately captures 3D patterns compared to other methods, as validated by the distribution of marker genes. The results are shown in Supplementary Fig.6a-e. We have revised the text on pages 8, and 22 to incorporate these changes.

Supplementary Figure 6. Method comparison on seven consecutive sections of the 3D hippocampal structure profiled by Slide-seq. **a** Visualization of 3D hippocampal structure, stacked from seven aligned consecutive slices. Each color represents one slice. **b** Spatial domains identified by SEDR, GraphST, PRECAST, STAligner, and stClinic, respectively. Each color represents a cluster. **c** UMAP visualization of features by SEDR, GraphST, PRECAST, STAligner, and stClinic, respectively. For each method, the left panel shows color-coded slices, while right panel displays color-coded clusters. **d** Comparison of the five methods (SEDR, GraphST, PRECAST, STAligner, and stClinic) based on cluster separation (ASW),

slice mixing (iLISI), and simultaneous cluster separation and slice mixing (F1 score). **e** Mean expression levels of marker genes across spatial clusters identified by SEDR, GraphST, PRECAST, STAligner, and stClinic, with mean expressions scaled by columns. Source data are provided as a Source Data file.

8 Rodrigues, S. G. *et al.* Slide-seq: A scalable technology for measuring genome-wide expression at high spatial resolution. *Science* **363**, 1463-1467 (2019).

Comment #5: Importantly, to determine if a cluster from one slice of a sample is identical or highly similar to another slice of the same sample or a different sample, the statistical assessment shown in Fig. 2E is insufficient. The morphological and histological similarity, as well as transcriptomic similarity, should also be evaluated to ensure that the batch correction was not under-corrected or overcorrected.

Response: We agree with the reviewer. In light of the comment, we have introduced novel computational methods to assess both histological and transcriptomic similarities between spots within the same clusters across different slices:

1. Morphological and histological similarities. We employed CONCH ⁹, a foundation model pretrained on extensive histology data, to extract histological features from spots. These features were projected into a shared space across different slices, and cosine similarity was calculated to assess the histological similarity of spots within the same clusters.

2. Transcriptomic similarity. Given the challenges of directly measuring transcriptomic similarity due to batch effects, we focused on known layer-specific genes that should ideally be over-expressed within specific clusters rather than across multiple clusters. We used the Gini index (GI) to evaluate the expression inequality of these layer-specific genes across clusters, providing a measure of transcriptional similarity between spots across slices.

Our analysis revealed that both stClinic and SEDR perform comparably in terms of histological similarity within clusters, outperforming other methods. However, stClinic achieves the highest GI score, indicating superior transcriptional alignment. These findings, combined with the statistical assessment in Fig.2E, demonstrated that stClinic effectively

aligns identical clusters across different slices. The detailed results for the comment are shown in Supplementary Fig.5b. In light of the comment, we have added the related text on page 7.

Supplementary Figure 5. Method comparison on 12 slices of the human DLPFC dataset.

a The Spatial domains identified by SEDR, PRECAST, STAligner, stClinic_fix, and stClinic. ARI scores for each method on each slice are displayed above the figure, with the highest values highlighted in red. **b** Comparison of the five methods (SEDR, PRECAST, STAligner, stClinic_fix, and stClinic) in terms of clustering accuracy (ASW), simultaneous cluster separation and slice mixing (F1 score), histological similarity, and marker gene specificity (GI score), illustrated by violin plots and three bar plots, respectively. **c** Bar chart showing slice compositions within each cluster predicted by each method, with annotations provided for comparison. **d** Spatial expression levels of marker genes in Layer 1 (*AQP4*) and Layer 2 (*HPCALI*) on 12 slices. Source data are provided as a Source Data file.

9 Lu, M. Y. *et al.* A visual-language foundation model for computational pathology. *Nature Medicine* **30**, 863-874, doi:10.1038/s41591-024-02856-4 (2024).

Comment #6: i) From Supplementary Figure 1a, stClinic did not outperform PRECAST and some other methods when considering ASW, F1, cLISI, and iLISI. Additionally, from Supplementary Figures 1b, 2a, 3a, and Figure 2d, the clustering separations were much clearer in PRECAST than in stClinic. ii) A bar plot to show the slice compositions in each cluster from each method could be a better way.

Response: We thank the reviewer for the comment. We have done the following for each issue. Specifically,

i) We appreciate the reviewer's observation and acknowledge that stClinic does not outperform all methods across every metric. Different metrics assess clustering, cluster mixing, and slice mixing from various perspectives: (1) ASW measures cluster separation, while the F1 score (derived from ASW) evaluates both cluster separation and slice mixing; and (2) cLISI and iLISI quantify mixing of clusters and slices in the neighborhood, whereas ARI and NMI assess clustering consistency with the ground truth.

While PRECAST achieves higher ASW and F1 scores than stClinic, indicating better cluster separation, stClinic excels in ARI and NMI scores, reflecting greater clustering

accuracy. For cLISI, SEDR and STAligner outperform stClinic in Sample 2, but stClinic surpasses all methods in Sample 1 and Sample 3. Its iLISI score is comparable to those of SEDR, Stitch3D, and STAligner, although all four methods underperform relative to GraphST and PRECAST. Importantly, stClinic demonstrates more accurate transcriptomic similarity of spots within clusters compared to other methods (refer to comment #5). This aligns with findings from the integrative analysis of seven consecutive sections of the 3D hippocampal structure, where stClinic, although not achieving the highest ASW, F1 score, or iLISI, demonstrates a more accurate marker gene distribution compared to other methods (refer to comment #4).

Overall, these findings highlight that stClinic effectively integrates identical clusters across slices within a batch-corrected feature space, enhancing the accuracy and reliability of multi-slice analysis. In light of the comment, we have added this clarification on pages 7, and 8.

ii) We agree with the reviewer. We have added bar plot to display the slice compositions in each cluster from each method. The results demonstrate that stClinic effectively mixes spots from the same layers across 12 slices, while other methods show over-corrected and disordered cortical layer structures. The relevant results are shown in Supplementary Fig.5c (refer to comment #5). In light of the comment, we have revised the related text on page 7.

Comment #7: To illustrate that stClinic can discern heterogeneous niches from complex disease tissues, this study analyzed ST data generated on breast cancer and hepatocellular carcinoma (Fig. 3). There are obvious limitations with the ST datasets utilized for analysis. The size of the tissues is overall small. The H&E images are low-resolution and somewhat blurry; most importantly, the pathology annotations are not at the spot level, the annotations are oversimplified and lack of essential granularity. This is clearly an issue. For example, large areas in panel a are labelled as “invasive cancer” while there are clearly non-neoplastic tissues within those regions; while in panel g, the complex tissues are labeled as “stroma” and “tumor/normal epithelium, which is overly simplified and not acceptable, as tumor/normal epithelium are mixed up in annotations and lack of essential granularity. It is therefore hard to trust the results in Fig. 3 and determine the spatial domains from which tool align better with

the pathology annotations. This is a major weakness of the study. The authors should select more appropriate ST datasets to ensure a robust assessment. Additionally, other approaches for spatial domain analysis such as Seurat, SpaGCN, and several recently published tools, should also be considered in the benchmarking.

Response: We appreciate the reviewer's comments regarding the limitations of the breast cancer and hepatocellular carcinoma ST datasets used in Fig.3. We acknowledge the concerns about the small tissue sizes, low-resolution H&E images, and oversimplified pathological annotations lacking spot-level granularity in these datasets ^{10,11}. Given these limitations, we have removed the analyses involving these two datasets from our manuscript.

To address these issues and ensure a more robust assessment, we conducted a comprehensive benchmarking analysis using two human Luminal B breast cancer samples with detailed tumor region annotations, each encompassing approximately 4000 spots. We compared the performance of stClinic against five other state-of-the-art methods, including Seurat and the latest method SLAT ¹². We decided not to include SpaGCN ¹³ in our comparison due to its specific focus on adjacent slices from a single tissue, as explained in the revised introduction.

By comparison, we found that (1) stClinic's spatial domains align more accurately with histological annotations, as indicated by the lowest hLISI score, reflecting greater homogeneity within neighborhoods of histological types. Notably, stClinic effectively delineates the boundaries of cluster 6 in the BAS1 sample; (2) despite the greater separation observed in the UMAP embeddings of PRECAST, stClinic consistently outperforms all methods in terms of inter-patient cluster consistency, as evidenced by its superior iLISI score and similar cell compositions across patients within the same cluster; (3) stClinic uniquely identifies cluster 10 (tumor edge) surrounding cluster 1 (tumor core), characterized by a significance presence of macrophages (e.g., *LYZ* and *CD74*), and memory B cells (e.g., *IGKC* and *IGHG3*). Cluster 10 also exhibits enhanced complement activation (e.g., *CIQA*, *CIQB*, and *C3*) and upregulated expression of HLA class II molecules (e.g., *HLA-DRB5*, *HLA-DPB1*, *HLA-DRA*, *HLA-DRB1*, and *HLA-E*), indicating a dual role in supporting tumor growth ¹⁴ and anti-tumor immune response. This highlights the complex interplay between inflammation and immune

modulation in tumor progression. These results are shown in Fig.3a-g and Supplementary Fig.7a and b.

In light of the comment, we have revised the manuscript accordingly, updating the relevant sections on pages 3, 8, 9, 22, 25, and 38 to reflect these enhancements. We believe these modifications significantly strengthen the robustness and validity of our study, providing clearer insights into the capability of stClinic to discern heterogeneous niches within complex disease tissues.

Fig.3. stClinic enables the identification of inter-tumoral niches in two human Luminal B breast cancer samples: IDC and BAS1. a H&E plots of IDC and BAS1 samples, with manual annotations indicating 12 and 11 tumor regions, respectively. **b** Spatial domains identified by SLAT, Seurat, SEDR, STAligner, PRECAST, and stClinic for both samples. **c** UMAP visualization of latent features for the two samples by SLAT, Seurat, SEDR, STAligner,

PRECAST, and stClinic, respectively. Colors in the top and bottom panels represent slices and clusters, respectively. **d** Comparison of slice mixing accuracy across six methods (SLAT, Seurat, SEDR, STAligner, PRECAST, and stClinic) based on histological annotation types (hLISI) and slice numbers (iLISI), as well as the evaluation of simultaneous cluster separation and slice mixing (F1 score). **e** Hierarchical clustering of cell type proportion ranks within each cluster for both samples. **f** Spatial distribution (left panel) and over-expressed genes of cluster 10 compared to cluster 1 (right panel). **g** Functional annotation of over-expressed genes in cluster 10 relative to cluster 1. Source data are provided as a Source Data file.

Supplementary Figure 7. a Boxplot showing expression levels of over-expressed genes (i.e., *HLA-E*, *HLA-DRB5*, *HLA-DPB1*, *C3*, *IGLC2*, *IGHG4*, *IGHG1*, *IGHA1*, and *IGLC1*) in cluster 10 compared to cluster 1. For each boxplot, the center line, box limits and whiskers separately indicate the median, upper and lower quartiles and $1.5 \times$ interquartile range. **b** Boxplot comparing the proportions of three cell types (including memory B cells, macrophages, and CD4 + T cells) in cluster 10 versus cluster 1 across IDC and BAS1 samples. Source data are provided as a Source Data file.

- 11 Liu, W. *et al.* Probabilistic embedding, clustering, and alignment for integrating spatial transcriptomics data with PRECAST. *Nat Commun* **14**, 296 (2023).
- 12 Xia, C.-R., Cao, Z.-J., Tu, X.-M. & Gao, G. Spatial-linked alignment tool (SLAT) for aligning heterogenous slices. *Nat Commun* **14**, 7236 (2023).
- 13 Hu, J. *et al.* SpaGCN: Integrating gene expression, spatial location and histology to identify spatial domains and spatially variable genes by graph convolutional network. *Nat Methods* **18**, 1342-1351, doi:10.1038/s41592-021-01255-8 (2021).
- 14 Roumenina, L. T. *et al.* Tumor cells hijack macrophage-produced complement C1q to promote tumor growth. *Cancer immunology research* **7**, 1091-1105 (2019).

Comment #8: In Fig. 3d, some clusters, such as clusters 1, 3, 7, showed very inconsistent cell compositions between the two samples (CID4465 and CID44971). In Fig. 3i, cluster 9 has multiple cell components and its functions are not clear. It is a bit confusing why the author specifically emphasizes this cluster.

Response: We thank the reviewer for the comment. In light of the comment #7, we have decided to remove the analysis of the breast cancer and hepatocellular carcinoma datasets, including the figures and discussion related to Fig.3d and Fig.3i. As such, the concerns regarding compositions and the emphasis on cluster 9 are no longer applicable to the revised manuscript. We have updated the manuscript accordingly.

Comment #9: i) Figure 4a is confusing. If the risk group is defined based on survival time, the survival difference will surely be significant. If so, this analysis is not really useful. ii) Additionally, it seems that the definitions of high vs. low risk are different in Figs. 4a and 4f. This is a bit confusing as well. The author should state this clearly in the results or use different naming to distinguish between them.

Response: We thank the reviewer for the insightful comment. We have addressed each issue as follows. Specifically,

i) Fig.4a shows the Kaplan-Meier survival curves for 43 breast cancer slices, with groups stratified into low-risk and high-risk based on the median overall survival time. While the significant survival difference ($p = 1.662 \times 10^{-8}$) is expected given the risk definition, this

analysis serves as a preliminary step to verify that the data can effectively distinguish survival associated clusters. Additionally, this survival data is further used to evaluate the predictive power of different clusters, highlighting their clinical relevance. In light of the comment, we have revised both Fig.4a and the related text on pages 9, and 39 for clarity.

ii) Fig.4f displays the Kaplan-Meier survival curves based on the median hazard ratio predicted by stClinic, where a low hazard ratio indicates longer survival. We acknowledge the potential confusion between the two definitions of risk groups. In light of the comment, we have revised Fig.4f and the related text on page 39 to clearly differentiate the two definitions.

Fig.4. stClinic evaluates the malignancy level of niches on the 43 TNBC slices. a Kaplan-Meier survival curves for 43 breast cancer slices, with slices stratified into low-risk and high-

risk groups based on their median overall survival time. The p -value was computed using the log-rank test. **b** UMAP visualization of the latent features by stClinic on 43 slices, where the colors in the left and right panels indicate slices and clusters, respectively. **c** Heatmap showing the cell type proportions on spatial domains by stClinic. **d** H&E plot of slices 15 and 26. Slice 15 was annotated with fibrosis, necrosis, tumor, and dense lymphoid infiltration regions, while slice 26 was annotated with fibrosis and tumor regions. **e** Spatial domains identified by stClinic on slices 15 and 26. **f** Kaplan-Meier survival curves for 43 breast cancer slices, with slices classified into low-risk and high-risk groups based on their median hazard ratio predicted by stClinic. **g** Bar plot displaying the weights of cluster in prognosis by stClinic. **h** Heatmap depicting the average gene expression of the top 30 over-expressed genes for clusters 10 and 6. **i** Overall survival rate of patients with the low or high expression patterns of the top 30 over-expressed genes for clusters 10 (top panel) and 6 (bottom panel) in breast cancer data from TCGA by GEPIA2 ¹⁵. **j** Functional enrichment analysis of the over-expressed genes in cluster 10. **k** Heatmap showing the inferred large-scale CNVs by InferCNV ¹⁶ from cluster 10 compared with cluster 6. Red indicates high CNV levels, while blue indicates low CNV levels. Source data are provided as a Source Data file.

15 Tang, Z., Kang, B., Li, C., Chen, T. & Zhang, Z. GEPIA2: an enhanced web server for large-scale expression profiling and interactive analysis. *Nucleic Acids Res* **47**, W556-W560 (2019).

16 Patel, A. P. *et al.* Single-cell RNA-seq highlights intratumoral heterogeneity in primary glioblastoma. *Science* **344**, 1396-1401 (2014).

Comment #10: In Fig. 4, the integration results look promising. However, it appears that the dense lymphoid infiltrate in panel d was not identified as a unique cluster in panel e and seems to be mixed with fibrosis, which is a very different tissue structure. What is the likely reason for missing this unique structure, leading to the mixing up of these tissue structures with distinct expression profiles?

Response: We thank the reviewer for the insightful comment. Upon re-examining the integration results, we conducted a detailed comparison of clusters 7 and 11, specifically focusing on the dense lymphoid infiltrate and fibrosis regions. Estimating the spatial distribution of different cell types from scRNA-seq data ¹⁰ by GraphST ¹⁷, we found that both regions consist of a mix of T, stromal, and myeloid cells. This mixed cellular compositions likely contributed to stClinic’s merging of these clusters, rather than identifying them as separated entities. Notably, other methods also failed to identify a unique cluster within the lymphoid infiltrate region. To better understand the heterogeneity of these cell populations, we performed a more refined clustering analysis using gene activity scores from T cell, stromal cell, and myeloid cell markers, identifying three distinct sub-clusters. We validated these sub-clusters through gene enrichment analysis of the up-regulated genes in each using the DAVID tool (<https://david.ncifcrf.gov/tools.jsp>). Each sub-cluster exhibited unique function characteristics: sub-cluster 1, primarily located in the dense lymphoid infiltrate region, is associated with T cell activation, differentiation, and proliferation; sub-cluster 2 is involved in cell cycle regulation and shows overexpression of *MMP2*, indicating its role in degrading components of the extracellular matrix ¹⁸; and sub-cluster 3 is enriched in the HIF1 signaling pathway, with over-expression of *PRXL2A* ¹⁹, suggesting its involvement in regulating endothelial cells response to hypoxia. The results for the comments are shown in Supplementary Fig.14a-d. We have included the related text on page 10.

Supplementary Figure 14. Sub-clustering analysis of clusters 7 and 11 on slice 15 of the TNBC sample. **a** Clustering analysis of gene activity scores of marker genes for T cells (*CD8A*, *CD8B*, *CD3E*, *CD3D*, *CD3G*, and *CD4*), stromal cells (*COL1A2*, *COL3A1*, *PDGFRA*, *COL1A1*, *PDGFRB*, *ACTA2*, *CD34*, and *CD74*), and myeloid cells (*CD163*, *CD68*, *CD14*, and *CSF1R*) within dense lymphoid infiltrate and fibrosis regions. **b** Violin plots showing marker gene expression across three sub-clusters: *BATF*, *ADAMDEC1*, *PTK2B*, and *LYZ* for sub-cluster 1; *COLGALT2*, *PELI2*, *MMP2*, and *THY1* for sub-cluster 2; and *PRXL2A*, *NES*, *APOD*, and *CD24* for sub-cluster 3. **c** Spatial distributions of clusters 7 and 11 along with the three

identified sub-clusters. **d** Functional annotation of over-expressed genes in each sub-cluster using DAVID (<https://david.ncifcrf.gov/tools.jsp>), where color represents the $-67810(\log_{10}(\text{value}))$ and size represents the gene count. Source data are provided as a Source Data file.

- 10 Wu, S. Z. *et al.* A single-cell and spatially resolved atlas of human breast cancers. *Nature genetics* **53**, 1334-1347 (2021).
- 17 Long, Y. *et al.* Spatially informed clustering, integration, and deconvolution of spatial transcriptomics with GraphST. *Nat Commun* **14**, 1155 (2023).
- 18 Jezierska, A. & Motyl, T. Matrix metalloproteinase-2 involvement in breast cancer progression: a mini-review. *Med Sci Monit* **15**, Ra32-40 (2009).
- 19 Chen, Y.-F. *et al.* miR-125b suppresses oral oncogenicity by targeting the anti-oxidative gene PRXL2A. *Redox Biology* **22**, 101140, doi:<https://doi.org/10.1016/j.redox.2019.101140> (2019).

Comment #11: The findings of cluster 6 and cluster 10 are interesting. However, i) it is too general to summarize cluster 6 and cluster 10 as malignant and non-malignant niches since cluster 10 also contains non-malignant cells. ii) Given that cluster 6 showed high expression of multiple immune signals, it would be beneficial to identify the major cell types and TME structures it captures. Additionally, further analysis/discussions are needed to provide insights into how these findings may translate to patient survival.

Response: We thank the reviewer for the thoughtful comment. We have addressed each issue as follows. Specifically,

i) We agree with the reviewer's suggestion and have replaced the terms "malignant" and "non-malignant" with "tumor-promoting" and "tumor-suppressive" throughout the manuscript to provide a more precise description.

ii) In light of the comment, we conducted a detailed analysis to identify the major cell types and tumor microenvironment (TME) structures within cluster 6. Through cell type annotation and gene expression analysis, we found that (1) cluster 6 predominantly comprises CD8⁺ T cells, B cells, and dendritic cells (DCs), suggesting its crucial role in the tumor immune

response through mechanisms such as CD8⁺ T-mediated cytotoxicity, antigen presentation by DCs, and antibody production by B cells; and (2) survival analysis based on markers identified in cluster 6 shows that the immune-active profile of this cluster may serve as a prognostic biomarker for better outcomes in patients with stronger anti-tumor immune responses. The results are shown in Fig.4c and i (ref to #comment #9) and Supplementary Figs.16 a and b and 17. We have added the relevant text on pages 10, and 11.

Supplementary Figure 16. a Spatial distribution of nine different cell types on slices 15 and 26 of the TNBC dataset. **b** Violin plot showing the expression levels of marker genes for B cells (*CD79A*, and *MZB1*), plasma B cells (*IGHG3*, and *IGHG4*), memory B cells (*CD27*, and *FCRL5*), dendritic cells (DCs) (*IRF7*, and *TLR7*), CD8⁺ T cells (*CD8A*, and *CD8B*), and cytotoxic T cells (*GNLY*, and *NKG7*) across cluster 6 and other groups.

Supplementary Figure 17. Functional annotation of over-expressed genes in cluster 6 by stClinic on the TNBC dataset.

Comment #12: For lymphocyte aggregates in the BC sample, the gene expression was diffused across the whole slide, especially IGHG1 and IGKC. Lymphocyte aggregates cannot be directly revealed from these genes. Are there any other more distinguishable markers? Please also provide zoom-in H&E images for these regions in parallel in Supplementary Figure 5c.

Response: We thank the reviewer for the comment. In light of the comment #7, we have decided to remove the analysis of the breast cancer samples, including the figures and discussion related to lymphocyte aggregates, as well as Supplementary Figure 5c. The manuscript has been updated accordingly.

Comment #13: i) In Figure 5i, CRC liver metastasis and liver cancer are substantially different. Please further validate the signature in CRC cohorts. ii) It would also be helpful to see if the signature can predict metastasis of CRC in independent cohorts.

Response: We appreciate the reviewer's insightful comments. We have done the following for each issue. Specifically,

i) We have conducted disease-free survival (DFS) analysis using CRC data from the TCGA database to assess the association between 10-gene signature and cancer recurrence. Our findings indicated that the 10-gene signature is significantly correlated with DFS, suggesting that its potential as a biomarker for predicting disease recurrence or death. The updated results are shown in Fig.5i. In light of the comment, we have revised the related text accordingly on pages 12, 41, and 42.

ii) We have trained a random forest model using the 10-gene signature with five-fold cross-validation, utilizing 90% of the TCGA data for training and 10% for testing, to distinguish between CRC stages I-III and stage IV (metastatic) cases. The model achieved an accuracy of 0.87 and an AUC score of 0.74 on the test data. To further validate the robustness of the signature, we conducted 1,000 random selections of 10-genes a pool of ~60,000 genes and repeated the same training strategy. Our results showed that the AUC score for the 10-gene signature is significantly higher than those of the random selections (p – value = 0.049), indicating its strong predictive power for CRC metastasis. In light of the comment, we have revised the related text accordingly on page 12.

[Figure Redacted]

- 15 Tang, Z., Kang, B., Li, C., Chen, T. & Zhang, Z. GEPIA2: an enhanced web server for large-scale expression profiling and interactive analysis. *Nucleic Acids Res* **47**, W556-W560 (2019).

Comment #14: The conclusion from Figure 5 that cluster 1 may represent the initial cell populations infiltrating normal liver tissue does not have enough evidence to be supported, and the definition of metastatic seeding clone will need genomic-level evidence.

Response: We appreciate the reviewer for pointing out the limitations of our conclusion. We acknowledge that the current data do not provide sufficient evidence to definitively support this interpretation, and we agree that identifying metastatic seeding clones requires genomic-level validation, such as mutational or lineage-tracing analyses. Accordingly, we have revised the discussion in the manuscript to clarify that while cluster 1 may suggest a potential role in early tissue infiltration, this remains a hypothesis that requires further genomic data for confirmation. In light of the comment, we have revised the related text on page 12.

Comment #15: In Figure 5g, the signals of some markers are diffused, such as FCGR3A and proliferative markers. Additionally, some markers, like SPP1, are not specifically enriched in the cluster 1 region. Given this, it is challenging to assert that SPP1+ myeloid cells may be uniquely enriched and play a role in this cluster. The analysis of the functions of this cluster needs significant improvement.

Response: Yes, we have re-analyzed the gene expression data in cluster 1 to provide a more detailed characterization of SPP1+ myeloid cells in this region. Our finding that SPP1+ myeloid cells uniquely over-express *MTRNR2L12* aligns with a previous study²⁰, which

reported that metastatic *SPP1+* macrophages exhibit higher *MTRNR2L12* expression compared to those in normal liver tissue. *MTRNR2L12*, a member of the humanin family known to inhibit apoptosis²¹, may support the survival and function of *SPP1+* macrophages within the tumor microenvironment, particularly in the context of metastasis. In light of the comment, we have revised Figure 5h (in the revised version, ref to comment #13) and updated the corresponding text on pages 2, 5, 12, 15, 41, and 42.

20 Sathe, A. *et al.* Colorectal cancer metastases in the liver establish immunosuppressive spatial networking between tumor-associated SPP1+ macrophages and fibroblasts. *Clinical Cancer Research* **29**, 244-260 (2023).

21 Morris, D. L., Johnson, S., Bleck, C. K., Lee, D.-Y. & Tjandra, N. Humanin selectively prevents the activation of pro-apoptotic protein BID by sequestering it into fibers. *Journal of Biological Chemistry* **295**, 18226-18238 (2020).

Comment #16: Lastly, i) for the label transfer function of stClinic, it would be beneficial to show an example from tumor samples. Tumors generally have a higher degree of heterogeneity, making the label transfer task more challenging. However, demonstrating this is very important for the application. Benchmarking with other published software is needed. ii) For multi-omics data integration, it might be helpful to also benchmark with MaxFuse, a recently published software aimed at diagonal integration.

Response: We thank the reviewer for the valuable suggestions, and have done the following for each issue. Specifically,

i) We performed an additional experiment using breast cancer samples, training stClinic on IDC and BAS1, and transferring labels to BAS2. For benchmarking, we compared stClinic with Seurat and Geneformer²², a single-cell RNA-seq foundation model that supports batch-effect correction and zero-shot learning. Both stClinic and Seurat successfully transfer labels, while Geneformer performs poorly. Notably, stClinic uniquely identifies cluster 9 surrounding cluster 6 in BAS1 and accurately transfers these labels to BAS2. In both BAS1 and BAS2, cluster 9 exhibits over-expression of basal cell markers (*KRT5*, *KRT14*, and *KRT17*)²³ and *SERPINA3*²⁴, and is involved in processes such as extracellular matrix remodeling, collagen

metabolism, cell migration, tissue repair, inflammatory response, and the PI3K-Akt signaling pathway, highlighting its critical role in promoting tumor cell invasion.

Additionally, in our analysis of the DLPFC sample, stClinic accurately classifies ~70% of spots, significantly outperforming Seurat and Geneformer, which achieve only 16~18% accuracy. These findings underscore stClinic's superior generalizability and its capability to transfer knowledge, thereby enhancing our understanding of complex biological systems. Relevant results are presented in Fig.3a-c, Fig.6b-i, and Supplementary Fig.26a-d. In light of the comment, we have incorporated the related text on pages 13, 22, 43, and 44.

ii) We have also benchmarked MaxFuse²⁵ for the multi-omics data integration of RNA-seq (from Stereo-seq)³ and ATAC-seq (from spatial ATAC-seq)²⁶ on mouse embryo slices. stClinic demonstrates superior alignment between RNA-seq and ATAC-seq, with transferred ATAC-seq labels from RNA-seq data showing greater consistency with known marker genes. The results are presented in Fig.7c-e and Supplementary Fig.27f. In light of the comment, we have revised the related text on pages 14, 45, and 46.

Fig.3. stClinic enables the identification of inter-tumoral niches in two human Luminal B breast cancer samples: IDC and BAS1. **a** H&E plots of IDC and BAS1 samples, with manual annotations indicating 12 and 11 tumor regions, respectively. **b** Spatial domains identified by SLAT, Seurat, SEDR, STAligner, PRECAST, and stClinic for both samples. **c** UMAP visualization of latent features for the two samples by SLAT, Seurat, SEDR, STAligner, PRECAST, and stClinic, respectively. Colors in the top and bottom panels represent slices and clusters, respectively. **d** Comparison of slice mixing accuracy across six methods (SLAT, Seurat, SEDR, STAligner, PRECAST, and stClinic) based on histological annotation types (hLISI) and slice numbers (iLISI), as well as the evaluation of simultaneous cluster separation and slice mixing (F1 score). **e** Hierarchical clustering of cell type proportion ranks within each cluster for both samples. **f** Spatial distribution (left panel) and over-expressed genes of cluster 10 compared to cluster 1 (right panel). **g** Functional annotation of over-expressed genes in cluster 10 relative to cluster 1. Source data are provided as a Source Data file.

[Figure Redacted]

Supplementary Figure 26. Evaluation of zero-shot learning on the human DLPFC and three Luminal B breast cancer samples. a Heatmap showing the proportion of interaction between each predicted cluster (by Geneformer, Seurat, and stClinic) and annotated cluster on slice 151676, compared to each annotated cluster. **b** Heatmap displaying the proportion of interaction between each predicted cluster (by Geneformer, Seurat, and stClinic) and annotated cluster on slice 151507, compared to each annotated cluster. **c** UMAP visualization of latent features by Geneformer and Seurat across three slices (IDC, BAS1, and BAS2). IDC and BAS1

serve as the reference set, while BAS2 is treated as the query sample. Spot labels in BAS2 are transferred from the reference set (right panel). **d** Spatial domains identified by Geneformer and Seurat on IDC and BAS1 samples. The results of Seurat are identical to those shown in **Fig.3b** and **c**. Source data are provided as a Source Data file.

Fig.7. stClinic improves the detection of finer structure by integrating spatial multi-omics data from the same and different slides. a stClinic learns joint features by integrating latent features from multi-omics tools like MultiVi alongside spatial location data within dynamic graphs. **b** Leveraging aligned features from multi-omics tools like Seurat in a multi-slice integrative condition, stClinic employs the same strategy as **a** to learn the final features. **c** Manual annotation of mouse brain coronal section, and spatial domains identified by single modality (RNA or ATAC), CellCharter, and stClinic. **d** Spatial ATAC levels of marker genes for cluster 1 (*Pde10a*), cluster 3 (*Cux2*), cluster 8 (*Mbp*), cluster 12 (*Dlx1*), and cluster 13 (*Drd3*), for mouse brain coronal section data denoised by stClinic. **e** Cell type distribution of brain, connective tissue, head mesenchyme, heart, liver, lung primordium, mesenchyme, spinal cord, and surface ectoderm on mouse embryo E10.5 tissue profiled by Stereo-seq. **f** Transferred cell type distribution on mouse embryo E11 tissue profiled by spatial ATAC-seq using MaxFuse, GLUE, SLAT, Seurat, and stClinic, respectively. **g** Spatial ATAC levels of marker genes for brain (*Sox1ot*), spinal cord (*Cntnap5b*), connective tissue (*Dnm3os*), and heart (*Tnn2*), for mouse embryo E11 data denoised by stClinic. Source data are provided as a Source Data file.

Supplementary Figure 27. Method comparison on spatial multi-omics dataset from the same slice and different slices. **a** UMAP visualization of the features by methods for single modality (RNA and ATAC), multi-modalities (multiVI, CellCharter, and stClinic), on mouse brain coronal section, respectively. Each color indicates one cluster. **b** Spatial domain identified by multiVI on mouse brain coronal section. **c** Spatial expression of marker genes for cluster 1 (*Pde10a*), cluster 3 (*Cux2*), cluster 7 (*Spock3*), cluster 8 (*Mbp*), cluster 12 (*Dlx1*), cluster 13

(*Drd3*), on mouse brain coronal section data denoised by stClinic. **d** Spatial ATAC levels of marker genes for cluster 7 (*Spock3*) on mouse brain coronal section data denoised by stClinic. **e** Violin plot displaying the gene expression (left panel) and ATAC levels (right panel) of marker genes (*Pde10a*, *Mef2c*, *Cux2*, *Neurod6*, *Spock3*, *Mbp*, *Sox10*, *Dlx1*, and *Drd3*) in each cluster by stClinic, respectively. **f** UMAP visualization of the latent features by MaxFuse, GLUE, SLAT, Seurat, and stClinic on mouse embryo sample. In the top and bottom panels, the colors represent the modality and cluster, respectively. Source data are provided as a Source Data file.

- 3 Chen, A. *et al.* Spatiotemporal transcriptomic atlas of mouse organogenesis using DNA nanoball-patterned arrays. *Cell* **185**, 1777-1792. e1721 (2022).
- 22 Theodoris, C. V. *et al.* Transfer learning enables predictions in network biology. *Nature* **618**, 616-624, doi:10.1038/s41586-023-06139-9 (2023).
- 23 Ren, Z. *et al.* Redox signalling regulates breast cancer metastasis via phenotypic and metabolic reprogramming due to p63 activation by HIF1 α . *Brit J Cancer* **130**, 908-924, doi:10.1038/s41416-023-02522-5 (2024).
- 24 Zhang, Y. *et al.* Overexpression of SERPINA3 promotes tumor invasion and migration, epithelial-mesenchymal-transition in triple-negative breast cancer cells. *Breast Cancer* **28**, 859-873 (2021).
- 25 Chen, S. *et al.* Integration of spatial and single-cell data across modalities with weakly linked features. *Nature Biotechnology* **42**, 1096-1106, doi:10.1038/s41587-023-01935-0 (2024).
- 26 Deng, Y. *et al.* Spatial profiling of chromatin accessibility in mouse and human tissues. *Nature* **609**, 375-383 (2022).

Comment #17: In page2, lines 22-23, more related studies should be listed to support why certain cellular niche could be clinically relevant. What makes studying cellular niches more important compared to studying individual TME cell states alone?

Response: Yes, we agree with the reviewer's suggestion. We have revised the text as follows: These complex interactions—such as those between tumor-associated macrophages, cancer-associated fibroblasts, and immune cells—form a dynamic network that significantly influences clinical outcomes, including tumor stage, grade, prognosis, and treatment response ²⁷⁻²⁹. Studying cellular niches, rather than focusing solely on individual cell-states, is crucial for capturing the collective behaviors and emergent properties of the tumor ecosystem, which provide a more comprehensive understanding of cancer dynamics and pave the way for precision medicine ²⁹⁻³¹. In light of the comment, we have revised the text to include additional studies that support the clinical relevance of cellular niches on pages 2 and 3.

27 Roma-Rodrigues, C., Mendes, R., Baptista, P. V. & Fernandes, A. R. Targeting tumor microenvironment for cancer therapy. *Int J Mol Sci* **20**, 840 (2019).

28 Zhang, Y. *et al.* Single-cell characterization of infiltrating T cells identifies novel targets for gallbladder cancer immunotherapy. *Cancer Letters* **586**, 216675, doi:<https://doi.org/10.1016/j.canlet.2024.216675> (2024).

29 Bagaev, A. *et al.* Conserved pan-cancer microenvironment subtypes predict response to immunotherapy. *Cancer cell* **39**, 845-865. e847 (2021).

30 Barcellos-Hoff, M. H., Lyden, D. & Wang, T. C. The evolution of the cancer niche during multistage carcinogenesis. *Nature Reviews Cancer* **13**, 511-518 (2013).

31 Quail, D. F. & Joyce, J. A. Microenvironmental regulation of tumor progression and metastasis. *Nature medicine* **19**, 1423-1437 (2013).

Comment #18: In page 3, lines 12-14, the author listed 3 published computational approaches but only pointed out the limitations of PASTE. What about the other two? Please add a bit more discussion on their limitations. Additionally, provide a definition of homogeneous/heterogeneous slices integration for the readers' understanding. How do they differ from each other in concept, and how do they differ from (iii)?

Response: Yes, we have revised the text on page 3 as suggested to clearly outline the limitations of the current approaches, provide detailed definitions of homogeneous and

heterogeneous slice integration, and clarify how they differ from (iii) to enhance the readers' understanding. The revised text is as follows:

(i) integration of SRT data from adjacent slices of a tissue (homogeneous integration): SpaGCN¹³ identifies spatial domains from multiple slices with coherent gene expression and histology but requires manual coordinate modification. PASTE^{32,33} aligns or integrates adjacent slices using optimal transport theory, followed by STitch3D³⁴ and GraphST³⁵, which construct unified neighbor graphs with three-dimensional (3D) spatial locations inferred by PASTE and apply GNNs for integration and spatial domain identification. However, the linear alignments in PASTE, and its derivatives STitch3D and GraphST, struggle to capture heterogeneity within the TME across diverse slices.

() integration of SRT data from slices across diverse tissues (heterogeneous integration): SEDR³⁶ combines autoencoder and GNN to integrate gene expression and spatial location for spatial domain identification, while PRECAST³⁷ performs dimension reduction and spatial clustering with straightforward projections. STAligner³⁸ integrates graph attention autoencoder and spot triplets to identify shared and specific spatial domains across diverse SRT datasets. Yet, the reliance on spot relations across diverse slices may limit their ability to accurately dissect shared and specific niches across heterogeneous patients.

(i) integration of spatial multi-omics data from the same or different slices: CellCharter³⁹ and SLAT¹² preprocess multi-omics data using scVI⁴⁰ and GLUE⁴¹, followed by graph modeling to learn shared features.

12 Xia, C.-R., Cao, Z.-J., Tu, X.-M. & Gao, G. Spatial-linked alignment tool (SLAT) for aligning heterogenous slices. *Nat Commun* **14**, 7236 (2023).

13 Hu, J. *et al.* SpaGCN: Integrating gene expression, spatial location and histology to identify spatial domains and spatially variable genes by graph convolutional network. *Nat Methods* **18**, 1342-1351, doi:10.1038/s41592-021-01255-8 (2021).

32 Zeira, R., Land, M., Strzalkowski, A. & Raphael, B. J. Alignment and integration of spatial transcriptomics data. *Nature Methods* **19**, 567-575, doi:10.1038/s41592-022-01459-6 (2022).

- 33 Xinhao, L., Ron, Z. & Benjamin, J. R. PASTE2: Partial Alignment of Multi-slice Spatially Resolved Transcriptomics Data. *bioRxiv*, 2023.2001.2008.523162, doi:10.1101/2023.01.08.523162 (2023).
- 34 Wang, G. *et al.* Construction of a 3D whole organism spatial atlas by joint modelling of multiple slices with deep neural networks. *Nat Mach Intell* **5**, 1200-1213 (2023).
- 35 Long, Y. *et al.* Spatially informed clustering, integration, and deconvolution of spatial transcriptomics with GraphST. *Nature Communications* **14**, 1155, doi:10.1038/s41467-023-36796-3 (2023).
- 36 Huazhu, F. *et al.* Unsupervised Spatially Embedded Deep Representation of Spatial Transcriptomics. *bioRxiv*, 2021.2006.2015.448542, doi:10.1101/2021.06.15.448542 (2021).
- 37 Liu, W. *et al.* Probabilistic embedding, clustering, and alignment for integrating spatial transcriptomics data with PRECAST. *Nature Communications* **14**, 296, doi:10.1038/s41467-023-35947-w (2023).
- 38 Zhou, X., Dong, K. & Zhang, S. Integrating spatial transcriptomics data across different conditions, technologies and developmental stages. *Nature Computational Science* **3**, 894-906 (2023).
- 39 Varrone, M., Tavernari, D., Santamaria-Martínez, A., Walsh, L. A. & Ciriello, G. CellCharter reveals spatial cell niches associated with tissue remodeling and cell plasticity. *Nature Genetics*, 1-11 (2023).
- 40 Lopez, R., Regier, J., Cole, M. B., Jordan, M. I. & Yosef, N. Deep generative modeling for single-cell transcriptomics. *Nat Methods* **15**, 1053-1058 (2018).
- 41 Cao, Z.-J. & Gao, G. Multi-omics single-cell data integration and regulatory inference with graph-linked embedding. *Nature Biotechnology* **40**, 1458-1466 (2022).

Comment #19: i) For Fig. 1, the author needs to provide the definitions of vertical and diagonal integration in the corresponding text for the readers' understanding. ii) Fig. 1 is too crowded and hard for the readers to capture the key points. Please simplify it.

Response: We thank the reviewer for the suggestions, and have addressed each issue as follows:

i) We have incorporated the definitions of vertical and diagonal integration on page 34 to improve the readers’ understanding. Specifically, vertical integration aligns and integrates multi-omics data within the same slice, while diagonal integration does so across different slices.

ii) We have simplified Fig.1 to make it easier for readers to grasp the key points. In light of the comment, we have also revised the related text concerning Fig.1 on pages 5, 6, 17, 20, 21, and 34.

[Figure Redacted]

Comment #20: In Fig. 2d-f, the degree of cluster mixing and the number of clusters depend on the parameter settings of each tool.

Response: We agree with the reviewer’s comment. To ensure a fair comparison in Fig.2d-f, we selected parameters for each tool based on the pipelines outlined in the original publications or tutorials, as the DLPFC dataset was consistently used in those studies. In light of the comment, we have included Supplementary Table 2, detailing the pipelines, and added related text on page 6 for clarification.

Dataset	Method	Clustering method	Pipeline
DLPFC	SEDR	mclust	https://sedr.readthedocs.io/en/latest/Tutorial3_Batch_integration.html
	GraphST	mclust	https://deepst-tutorials.readthedocs.io/en/latest/Tutorial%20Vertical%20Integration.html
	Stitch3D	Gaussian Mixture Model	https://stitch3d-tutorial.readthedocs.io/en/latest/tutorials/DLPFC/STitch3D_DLPFC.html
	PRECAST	Built-in spatial factor model	https://feiyong.github.io/PRECAST/articles/PRECAST.DLPFC4.html

	STAligner	mclust	https://staligner.readthedocs.io/en/latest/Tutorial_DLPFC.html
	stClinic	mclust	https://github.com/cmzuo11/stClinic/
Tumor samples	SLAT	Leiden	https://slat.readthedocs.io/en/latest/tutorials/cross_technology.html
	Seurat	Louvain	https://satijalab.org/seurat/articles/spatial_vignette
	SEDR	mclust	https://sedr.readthedocs.io/en/latest/Tutorial3_Batch_integration.html
	STAligner	Louvain	https://staligner.readthedocs.io/en/latest/Tutorial_embryo.html
	PRECAST	Built-in spatial factor model	https://feivoung.github.io/PRECAST/articles/PRECAST.BreastCancer.html
	stClinic	Louvain	https://github.com/cmzuo11/stClinic/

Supplementary Table 2. Summary of clustering methods used for comparison across different datasets.

Comment #21: In Fig. 3, when looking at the results in detail, cluster 4 of stclinic in panel c does not seem to match the locations, numbers, and sizes of lymphocyte aggregates in panel a.

Response: We thank the reviewer for the comment. In light of the comment #7, we have decided to remove the analysis of the breast cancer samples, including the Fig.3a and c. The manuscript has been updated accordingly.

Comment #22: i) In page 7, line 4-16, and page 8, lines 1-18, please cite the figure panels right after each point, instead of putting all the figure panels to the end. ii) The citation of Figure 3 panels is not in order. Also, Figure 3a and g showed pathology annotations in different ways, please make them consistent.

Response: We thank the reviewer for the valuable comment. We have addressed each concern as follows. Specifically,

i) We have revised the manuscript to cite the figure panels immediately after each relevant point in the text, rather than grouping them all at the end. This adjustment has been made to improve clarity, as suggested.

ii) In light of the comment #7, we have decided to remove the analysis of the breast cancer samples, including the Fig.3a and g.

Comment #23: Please provide the DEG list of clusters defined by stClinic, and add the key genes shown in Fig. 3e to the dotplot in Supplementary Figure 6.

Response: In light of the comment #7, we have decided to remove the analysis of the breast cancer samples, including the Fig.3e and Supplementary Figure 6.

Comment #24: the webpage lacks detailed documentation for uses to run the tool and interpret the results.

Response: We agree with the reviewer. In light of the comment, we have updated our webpage to include comprehensive step-by-step instructions on installing and running stClinic, along with example workflows and detailed guidelines on interpreting the output data. You can access the improved documentation at the following link: <https://github.com/cmzuo11/stClinic>.

Reviewer #3: (Early-Career Researcher co-reviewer)

Comment #1: I co-reviewed this manuscript with one of the reviewers who provided the listed reports. This is part of the Nature Communications initiative to facilitate training in peer review and to provide appropriate recognition for Early Career Researchers who co-review manuscripts.

Response: Yes, we thank the reviewer for the comment and encouragement.

Reviewer #1:

Comment #1: All my comments have been addressed in the revised manuscript.

Response: We thank the reviewer for the comment and agreement.

Reviewer #2:

Comment #1: The authors have addressed most of the comments and made improvements to the manuscript. However, some key concerns remain unaddressed.

Response: We thank the reviewer for the comment and encouragement.

Comment #2: Original Comment #3: The performance of stClinic should be evaluated in additional samples and cancer types beyond those presented in the previous study (Ref #7). The tissue and annotations from Ref #7 are relatively simple and do not reflect the complexity of real-world tissue contexts. This limitation undermines the generalizability of the tool's performance.

Response: We thank the reviewer for the insightful comment. In light of the comment, we have conducted additional experiments on two breast cancer slices, BAS1 and BAS2, which include detailed pathological annotations with 20 groups to address the lack of granularity. These annotations ensure spot-level precision and reflect the heterogeneity of tumor tissues, providing a robust foundation for evaluating stClinic.

By benchmarking stClinic against seven recently published methods (SpaGCN, Seurat, SLAT, SEDR, STAligner, PRECAST, and BANKSY), we found that (i) while PRECAST achieves a higher F1 score, stClinic demonstrates superior ARI performance, exceeding PRECAST by more than 0.1. This highlights stClinic's ability to capture local details and maintain pairwise annotation consistency. Notably, stClinic uniquely detects tumor boundaries, such as cluster 17 (indicated by black arrows), which is characterized by overexpression of oxidative phosphorylation and glycolysis genes, potentially reflecting tumor cell invasion and metastasis¹ (Fig.3a-e and Supplementary Fig.7a); (ii) BANKSY achieves a slightly higher NMI

score than stClinic, reflecting its strength in capturing global categorical relationships. However, BANKSY fails to identify local small domains across slices, such as IDC_1 (outlined in black) and CIS_2 (outlined in red) (Fig.3a and c); (iii) for cLISI, BANKSY, SLAT, and SEDR outperform stClinic, while for iLISI, PRECAST and Seurat perform better than stClinic (Fig.3c); (iv) SpaGCN, despite its ability to identify spatial domains, struggles to align several common niches across slices due to its lack of batch effect correction capabilities (Fig.3a and b); and (v) stClinic demonstrates superior robustness to variations in the number of clusters, consistently maintaining overall high clustering and batch-effect correction performance in terms of F1 score, ARI, and NMI (Supplementary Fig.7b). Overall, these findings underscore stClinic's effectiveness in integrating multi-slice spatial transcriptomics data within a batch-corrected feature space, enabling accurate and reliable analysis of tumor heterogeneity. The results are shown in Fig.3a-e, and Supplementary Fig.7a, b. In light of the comment, we have added the related text on pages 3, 4, 8, 9, and 38.

Fig.3. stClinic enables the identification of intra-tumoral niches in two human Luminal B breast cancer slices: BAS1 and BAS2. a Manual pathological annotation of Hematoxylin and Eosin (H&E)-stained plots and clusters identified by SpaGCN, Seurat, SLAT, SEDR, STAligner, PRECAST, BANKSY, and stClinic. **b** UMAP visualization of latent features for the two slices generated by eight methods. Colors in the top and bottom panels represent slices and clusters, respectively. **c** Comparison of clustering accuracy (ARI and NMI) based on manual annotations, slice mixing accuracy using annotations (cLISI) and slice numbers (iLISI), and combined cluster separation and slice mixing performance (F1 score) across eight methods. **d** Over-expressed genes in cluster 17 compared to other clusters. **e** Functional annotation of over-expressed genes in cluster 17 relative to other clusters. Source data are provided as a Source Data file.

Supplementary Figure 7. Comparison of methods for the integrative analysis of BAS1 and BAS2 slices. a Over-expressed genes in cluster 17 compared to other clusters. **b** Performance evaluation of batch-effect correction (F1 score) and clustering accuracy (ARI and NMI) across varying numbers of clusters (12 to 20). Source data are provided as a Source Data file.

1 Liu, Y. M. *et al.* Combined Single-Cell and Spatial Transcriptomics Reveal the Metabolic Evolvement of Breast Cancer during Early Dissemination. *Adv Sci (Weinh)* **10**, e2205395, doi:10.1002/advs.202205395 (2023).

Comment #3: Original Comments #7 and #16: i) For the newly added breast cancer tissue, the pathology annotations lack essential granularity, making them unsuitable as ground truth for evaluating the performance of different tools. ii) Furthermore, the benchmarking is incomplete, as other highly ranked tools, such as BANKSY (Singhal et al., Nature Genetics 2024), have not been included. The authors removed the tool SpaGCN from the results; however, the rationale provided for its removal is unconvincing. iii) Additionally, the performance of these tools may be influenced by the resolution of clustering analysis (i.e., the number of clusters) and parameter settings.

Response: We thank the reviewer for the comment. We have done the following studies for each issue. Specifically,

i) We acknowledge that the pathological annotations for the newly added breast cancer tissue lack the granularity needed to serve as a definitive ground truth for evaluating tool performance. Providing precise pathological annotations for the VIDC slice is challenging due to the reliance on immunofluorescent staining without a corresponding H&E-stained image. To address this limitation, we have retained the VIDC analysis related to label transfer while noting its limitations and complemented it with detailed pathological annotations for BAS1 and BAS2, which include 20 groups and enable a more comprehensive comparison across different methods.

In light of the comment, we have performed comprehensive comparison between different methods on integrating BAS1 and BAS2. By analysis, we found that stClinic is able to capture local details and maintain pairwise consistency with annotation. The results are shown in Fig.3a-e and Supplementary Fig.7a, b (ref to comment #2). In addition, we benchmarked label transfer performance from VIDC and BAS1 to BAS2 against Seurat and Geneformer, showing that stClinic's labels align more closely with the annotations, as evidenced by higher ARI and NMI scores. The results are shown in Fig.6e-j, and the related text has been added on pages 3, 4, 8, 9, 13, 14, 38, 43, and 44.

ii) We appreciate the reviewer's comment. In light of the comment, we have included BANKSY² and SpaGCN³ in our benchmark comparison for the integrative analysis of BAS1 and BAS2, and also updated the introduction to include descriptions of both methods.

Additionally, we have revised the introduction of SpaGCN as follows: SpaGCN identifies spatial domains across multi-slices with coherent gene expression and histology but lacks batch-effect correction capabilities. This conclusion is supported by feedback from Jian Hu (<https://github.com/jianhuupenn/SpaGCN/issues/20>), and results from our benchmark experiments. The results of this comparison are presented in Fig.3a-c and Supplementary Fig.7b (ref to comment #2), and the related text has been added on pages 3, 4, 8, 9, and 38.

iii) We thank the reviewer for the insightful comment. For each tool, we followed the recommended pipeline by the authors to generate results. Additionally, we conducted a benchmarking comparison of different methods based on their performance in multi-slice integration and label transfer, focusing on F1 score, ARI, and NMI. It is important to note that cLISI and iLISI scores are independent of the number of clusters and were therefore excluded from this comparison.

In the integration of BAS1 and BAS2, we found that stClinic demonstrates greater robustness to variations in the number of clusters, consistently achieving overall high clustering and batch-effect correction performance as measured by F1 score, ARI, and NMI (Supplementary Fig.7b, referring to comment #2). Similarly, for the integration of VIDC and BAS1 with label transfer to BAS2, stClinic maintains consistently high clustering, batch-effect correction, and label transfer performance across varying clusters of numbers, as shown in Fig.6g, and Supplementary Figs.27a, 28. In light of the comment, we have included the detailed pipeline for each method in Supplementary Table 2, and added the related text on pages 9, 13, 43, and 44.

[Figure Redacted]

Supplementary Figure 27. Comparison of methods for the integration analysis of VIDC and BAS1 slices. **a** Evaluation of simultaneous cluster separation and slice mixing (F1 score) across varying numbers of clusters (12 to 20) for Geneformer, Seurat, and stClinic. **b** Slice mixing accuracy comparison of three methods based on histological annotation types (hLISI) and slice numbers (iLISI). **c** Hierarchical clustering of cell type proportion ranks within each cluster across the two slices. **d** Spatial distribution of clusters 10 and 1. **e** Boxplot of expression levels of markers of macrophages (*CD68* and *CD14*), CD4 + T cells (*CD4* and *IL7R*), memory B cells (*CD27*), HLA class II molecules (*HLA-DRB5*, *HLA-DPB1*, *HLA-DRA*, *HLA-DRB1*, and *HLA-E*), and complement activation (*CIQA* and *CIQB*) in cluster 10 versus cluster 1. $2.22e-16$ represents a very small numbers, effectively close to zero. **f** Boxplot comparing the

proportions of three cell types (macrophages, CD4 + T cells, and memory B cells) in cluster 10 versus cluster 1 across two slices. **g** Functional annotation of over-expressed genes in cluster 10 relative to cluster 1. For each boxplot of **e** and **f**, the center line, box limits and whiskers separately indicate the median, upper and lower quartiles and $1.5 \times$ interquartile range. Source data are provided as a Source Data file.

Supplementary Figure 28. Comparison of methods for the integration analysis of VIDC and BAS1 and label transfer to BAS2 across varying numbers of clusters (12 to 20) for Geneformer, Seurat, and stClinic. Source data are provided as a Source Data file.

Application	Method	Clustering method	Pipeline
Integrative analysis of the DLPFC dataset	SEDR	mclust	https://sedr.readthedocs.io/en/latest/Tutorial3
			Batch integration.html
	GraphST	mclust	https://deepst-
			tutorials.readthedocs.io/en/latest/Tutorial%205
			Vertical%20Integration.html
	Stitch3D	Gaussian Mixture Model	https://stitch3d-
			tutorial.readthedocs.io/en/latest/tutorials/DLPFC
			C/Stitch3D_DLPFC.html
	PRECAST	Built-in spatial factor model	https://feiyong.github.io/PRECAST/articles/P
			RECAST.DLPFC4.html
STAligner	mclust	https://staligner.readthedocs.io/en/latest/Tutori	
		al_DLPFC.html	
stClinic	mclust	https://github.com/cmzuo11/stClinic/	
Integrative analysis of tumor samples	SpaGCN	Louvain	https://github.com/jianhuupenn/SpaGCN/blob/
			master/tutorial/tutorial.ipynb
	SLAT	Leiden	https://slat.readthedocs.io/en/latest/tutorials/cro
			ss technology.html
	Seurat	Louvain	https://satijalab.org/seurat/articles/spatial_vign
			ette

	SEDR	mclust	https://sedr.readthedocs.io/en/latest/Tutorial3
			Batch integration.html
	STAligner	Louvain	https://staligner.readthedocs.io/en/latest/Tutorial_embryo.html
			al_embryo.html
	PRECAST	Built-in spatial factor model	https://feiyong.github.io/PRECAST/articles/PRECAST.BreastCancer.html
RECAST.BreastCancer.html			
BANKSY	Leiden	https://prabhakarlab.github.io/Banksy/articles/multi-sample.html	
		multi-sample.html	
stClinic	Louvain	https://github.com/cmzuo11/stClinic/	
Label Transfer	Geneformer	Louvain	https://geneformer.readthedocs.io/en/latest/getstarted.html and
			https://chanzuckerberg.github.io/cellxgene-census/notebooks/analysis_demo/comp_bio_geneformer_prediction.html
			neformer_prediction.html
			neformer_prediction.html
	Seurat	Louvain	https://satijalab.org/seurat/articles/integration_mapping
			mapping
	stClinic	Louvain	https://github.com/cmzuo11/stClinic/

Supplementary Table 2. Summary of clustering methods and pipelines used for comparisons across different experiments.

- 2 Singhal, V. *et al.* BANKSY unifies cell typing and tissue domain segmentation for scalable spatial omics data analysis. *Nature Genetics* **56**, 431-441, doi:10.1038/s41588-024-01664-3 (2024).

3 Hu, J. *et al.* SpaGCN: Integrating gene expression, spatial location and histology to identify spatial domains and spatially variable genes by graph convolutional network. *Nat Methods* **18**, 1342-1351, doi:10.1038/s41592-021-01255-8 (2021).

Comment #4: Marker Issues: The markers *LYZ* and *CD74* are not unique to macrophages, and similarly, *IGKC* and *IGHG3* are not unique markers for memory B cells. This issue raises concerns about the accuracy of cell type annotations.

Response: We appreciate the reviewer for pointing out this issue. To address this, we conducted additional analyses to validate the cell type annotations derived from a previous study ⁴. Specifically, we evaluated the distribution macrophages markers (*CD68* and *CD14*), and memory B cell marker (*CD27*) between clusters 10 and 1. The results demonstrated a significant enrichment of macrophages and memory B cells in cluster 10 compared to cluster 1, supporting the robustness of these annotations. The results are shown in Supplementary Fig.27e-g (ref to comment #3), and the corresponding text has been revised on page 13.

4 Wu, S. Z. *et al.* A single-cell and spatially resolved atlas of human breast cancers. *Nature genetics* **53**, 1334-1347 (2021).

Comment #5: CNVs Plot in Fig. 4 (Panel K): The copy number variation (CNV) plot appears very noisy. The authors should consider employing alternative methods for CNV inference to improve the clarity and reliability of this analysis.

Response: We thank the reviewer for the valuable comment. In light of the comment, we have re-inferred copy number variations (CNVs) using CopyKAT ⁵, and observed that higher copy number gains and losses in cluster 10 compared to cluster 6. Moreover, we applied inferCNV following the pipeline described in a previous study ⁶. The updated analysis produced clearer and more reliable results, further confirming the significantly elevated CNVs in cluster 10 relative to cluster 6. These revised results are now presented in Fig.4k and Supplementary Fig.17b-d, with corresponding updates to the text on pages 11, and 40.

Fig.4. stClinic evaluates the malignancy level of niches on the 43 TNBC slices. **a** Kaplan-Meier survival curves for 43 breast cancer slices, with slices stratified into low-risk and high-risk groups based on their median overall survival time. The p -value was computed using the log-rank test. **b** UMAP visualization of the latent features by stClinic on 43 slices, where the colors in the left and right panels indicate slices and clusters, respectively. **c** Heatmap showing the cell type proportions in spatial domains by stClinic. **d** H&E plot of slices 15 and 26. Slice 15 was annotated with fibrosis, necrosis, tumor, and dense lymphoid infiltration regions, while slice 26 was annotated with fibrosis and tumor regions. **e** Spatial domains identified by stClinic on slices 15 and 26. **f** Kaplan-Meier survival curves for 43 breast cancer slices, with slices classified into low-risk and high-risk groups based on their median hazard ratio predicted by stClinic. **g** Bar plot displaying the weights of cluster in prognosis by stClinic. **h** Heatmap

depicting the average gene expression of the top 30 over-expressed genes for clusters 10 and 6. **i** Overall survival rate of patients with the low or high expression patterns of the top 30 over-expressed genes for clusters 10 (top panel) and 6 (bottom panel) in breast cancer data from TCGA by GEPIA2 ⁷. **j** Functional enrichment analysis of the over-expressed genes in cluster 10. **k** Boxplot showing the total copy number gains and losses per spot in clusters 10 and 6, as inferred by CopyKAT ⁵. 2.22e-16 represents a very small numbers, effectively close to zero. For each boxplot, the center line, box limits and whiskers separately indicate the median, upper and lower quartiles and $1.5 \times$ interquartile range. Source data are provided as a Source Data file.

Supplementary Figure 17. **a** Functional annotation of over-expressed genes in cluster 6 identified by stClinic in the TNBC dataset. **b** Copy number variations (CNVs) in clusters 6 and 10, estimated using CopyKAT ⁵. **c** CNVs in clusters 6 and 10, estimated using inferCNV ⁶. **d**

Boxplot showing the total number of CNVs per spot for clusters 10 and cluster 6, as inferred by inferCNV. $2.22e-16$ represents a very small numbers, effectively close to zero. For each boxplot, the center line, box limits and whiskers separately indicate the median, upper and lower quartiles and $1.5 \times$ interquartile range. Source data are provided as a Source Data file.

- 5 Gao, R. *et al.* Delineating copy number and clonal substructure in human tumors from single-cell transcriptomes. *Nature biotechnology* **39**, 599-608 (2021).
- 6 Erickson, A. *et al.* Spatially resolved clonal copy number alterations in benign and malignant tissue. *Nature* **608**, 360-367, doi:10.1038/s41586-022-05023-2 (2022).
- 7 Tang, Z., Kang, B., Li, C., Chen, T. & Zhang, Z. GEPIA2: an enhanced web server for large-scale expression profiling and interactive analysis. *Nucleic Acids Res* **47**, W556-W560 (2019).

Comment #6: Original Comment #10: The response to this comment is unsatisfactory and highlights potential limitations of the tool. This issue should be revisited to ensure the tool's robustness.

Original Comment #10: In Fig. 4, the integration results look promising. However, it appears that the dense lymphoid infiltrate in panel d was not identified as a unique cluster in panel e and seems to be mixed with fibrosis, which is a very different tissue structure. What is the likely reason for missing this unique structure, leading to the mixing up of these tissue structures with distinct expression profiles?

Response: We thank the reviewer for the valuable comment. In light of the comment, we have performed sub-clustering analysis on the low-dimensional embeddings of spots from the dense lymphoid and fibrosis regions. Using graph-based models (SEDR, STAligner, and stClinic), we predicted three clusters and compared their performance.

Our analysis showed that stClinic and STAligner identify detailed structures, while SEDR performs less effectively, as shown in slice 15. Further examination revealed that perivascular-like (PVL), B, T, and myeloid cells are enriched across the three clusters. Both stClinic and STAligner detected two clusters enriched in T and B cells or myeloid cells. Notably, stClinic

shows significant PVL enrichment in one cluster compared to the other two, whereas STAligner displays PVL enrichment in one cluster relative to a single other cluster.

These results highlight stClinic’s advantage in dynamically aggregating neighborhood information, enabling more accurate identification of distinct niches compared to the fixed neighborhood approaches by STAligner and SEDR. The updated results are shown in Supplementary Fig.14a-c, and the corresponding text has been revised on page 10.

Supplementary Figure 14. Sub-clustering analysis of spots from the dense lymphoid and fibrosis regions in the human TNBC dataset. **a** Sub-clustering analysis of spot embeddings generated by three graph-based models: SEDR, STAligner, and stClinic. **b** Spatial distribution of three sub-clusters identified by three methods on slice 15. **c** Boxplots showing cell type proportions across the three sub-clusters by STAligner (top) and stClinic (bottom). 2.22e-16 represents a very small numbers, effectively close to zero. For each boxplot, the center line,

box limits and whiskers separately indicate the median, upper and lower quartiles and $1.5 \times$ interquartile range. Source data are provided as a Source Data file.

Comment #7: Original Comment #11: i) Naming clusters as “tumor-promoting” and “tumor-suppressive” is inappropriate without supporting functional data. ii) Additionally, the gene signatures used in Supplementary Figure 16a need further refinement to ensure their uniqueness and accuracy.

Response: We thank the reviewer for the comment. We have done the following analyses for each issue. Specifically,

i) We agree with the reviewer’s comment and have replaced the terms “tumor-promoting” and “tumor-suppressive” with “high-risk” and “low-risk” throughout the manuscript to provide a more accurate description.

ii) We appreciate the reviewer’s valuable comment. In light of the comment, we have refined the gene signatures for different cell types, ensuring their uniqueness by referencing findings from the previous study⁴. The updated results are presented in Supplementary Figure 16a.

Supplementary Figure 16. a Spatial feature plots of gene signature score for plasma B cells (*IGHG1*, *IGHG2*, *IGHG3*, and *IGHG4*), memory B cells (*CD27* and *FCRL5*), CD8 + T cells (*CD8A* and *CD8B*), cytotoxic T cells (*GNLY* and *NKG7*), CD4 + T cells (*IL7R*, *CCR7*, and *CD4*), DC (*IRF7* and *LAMP3*), Monocyte (*FCGR3A*), M1 Macrophage (*CXCL10*), M2 Macrophage (*EGR1* and *SIGLEC1*), Epithelial (*CDH1*, *KRT19*, *EPCAM*, and *CLDN7*), endothelial (*CDH5*, *PECAM1*, *VWF*, and *ENG*), CAF (*COL1A1*, *COL1A2*, *COL3A1*, *DCN*, and *MMP2*), PVL (*ACTA2* and *PDGFRB*), plasmablasts (*JCHAIN*) on slices 15 and 26 of the TNBC dataset. **b** Violin plot showing the expression levels of marker genes for B cells (*CD79A*,

and *MZB1*), plasma B cells (*IGHG3* and *IGHG4*), memory B cells (*CD27* and *FCRL5*), DC (*IRF7* and *TLR7*), CD8+ T cells (*CD8A*, and *CD8B*), and cytotoxic T cells (*GNLY*, and *NKG7*) across cluster 6 and other groups. Source data are provided as a Source Data file.

4 Wu, S. Z. *et al.* A single-cell and spatially resolved atlas of human breast cancers. *Nature genetics* **53**, 1334-1347 (2021).

Comment #8: Overall, while the authors have made progress in addressing many of the earlier comments, the issues outlined above require further attention to strengthen the rigor, robustness, and interpretability of the manuscript.

Response: We sincerely thank the reviewer for their valuable comments and constructive feedback. We have carefully addressed all the concerns raised and provided detailed responses to ensure the rigor, robustness, and interpretability of the manuscript. These revisions have significantly improved both the content and presentation, as outlined above.

Reviewer #3: (Early-Career Researcher co-reviewer)

Comment #1: I co-reviewed this manuscript with one of the reviewers who provided the listed reports. This is part of the Nature Communications initiative to facilitate training in peer review and to provide appropriate recognition for Early Career Researchers who co-review manuscripts.

Response: Yes, we thank the reviewer for the comment and encouragement.